# A Primal-Dual Approach to Bilevel Optimization with Multiple Inner Minima

## Abstract

Bilevel optimization has found extensive applications in modern machine learning problems such as hyperparameter optimization, neural architecture search, meta-learning, etc. While bilevel problems with a unique inner minimal point (e.g., where the inner function is strongly convex) are well understood, such a problem with multiple inner minimal points remains to be challenging and open. Existing algorithms designed for such a problem were applicable to restricted situations and do not come with a full guarantee of convergence. In this paper, we adopt a reformulation of bilevel optimization to constrained optimization, and solve the problem via a primal-dual bilevel optimization (PDBO) algorithm. PDBO not only addresses the multiple inner minima challenge, but also features fully first-order efficiency without involving second-order Hessian and Jacobian computations, as opposed to most existing gradient-based bilevel algorithms. We further characterize the convergence rate of PDBO, which serves as the first known non-asymptotic convergence guarantee for bilevel optimization with multiple inner minima. Our experiments demonstrate desired performance of the proposed approach.

## 1 Introduction

Bilevel optimization has received extensive attention recently due to its applications in a variety of modern machine learning problems. Typically, parameters handled by bilevel optimization are divided into two different types such as meta and base learners in few-shot meta-learning (Bertinetto et al., 2018; Rajeswaran et al., 2019), hyperparameters and model parameters training in automated hyperparameter tuning (Franceschi et al., 2018; Shaban et al., 2019), actors and critics in reinforcement learning (Konda & Tsitsiklis, 2000; Hong et al., 2020), and model architectures and weights in neural architecture search (Liu et al., 2018).

Mathematically, bilevel optimization captures intrinsic hierarchical structures in those machine learning models, and can be formulated into the following two-level problem:

$$\min_{x \in \mathcal{X}, y \in \mathcal{S}_x} f(x, y) \quad \text{with} \quad \mathcal{S}_x = \arg\min_{y \in \mathbb{R}^d} g(x, y), \tag{1}$$

where $f(x, y)$ and $g(x, y)$, the outer- and inner-level objective functions, are continuously differentiable, and the supports $\mathcal{X} \subseteq \mathbb{R}^p$ is convex, closed and bounded. For a fixed $x \in \mathcal{X}$, $\mathcal{S}_x$ is the set of all $y \in \mathbb{R}^d$ that yields the minimal value of $g(x, \cdot)$.

A broad collection of approaches have been proposed to solve the bilevel problem in eq. (1). Among them, gradient based algorithms have shown great effectiveness and efficiency in various deep learning applications, which include approximated implicit differentiation (AID) based methods (Domke, 2012; Pedregosa, 2016; Gould et al., 2016; Liao et al., 2018; Lorraine et al., 2020; Ji et al., 2021) and iterative differentiation (ITD) (or dynamic system) based methods (Maclaurin et al., 2015; Franceschi et al., 2017; Shaban et al., 2019; Grazzi et al., 2020b; Liu et al., 2020; 2021a). Many stochastic bilevel algorithms have been proposed recently via stochastic gradients (Ghadimi & Wang, 2018; Hong et al., 2020; Ji et al., 2021), and variance reduction (Yang et al., 2021) and momentum (Chen et al., 2021; Khanduri et al., 2021; Guo & Yang, 2021).

Most of these studies rely on the simplification that for each outer variable $x$, the inner-level problem has a **single** global minimal point. The studies for a more challenging scenario with **multiple** inner-level solutions (i.e., $\mathcal{S}_x$ has multiple elements) are rather limited. In fact, a counter example has been provided in Liu et al. (2020) to illustrate that simply applying algorithms designed for the single inner minima case will fail to optimize bilevel problems with multiple inner minima. Thus, bilevel problems with multiple inner minima deserve serious efforts of exploration. Recent studies (Liu et al., 2020; Li et al., 2020) proposed a gradient aggregation method and another study (Liu et al., 2021a) proposed a value-function-based method from a constrained optimization view to address the issue of multiple inner minima. However, all of these approaches take a *double-level* optimization structure, updating the outer variable $x$ after fully updating $y$ over the inner and outer functions, which could lose efficiency and cause difficulty in implementations. Further, these approaches have been provided with only the *asymptotic* convergence guarantee without characterization of the convergence rate.

*The focus of this paper is to develop a better-structured bilevel optimization algorithm, which handles the multiple inner minima challenge and comes with a finite-time convergence rate guarantee.*

### 1.1   Our Contributions

In this paper, we adopt a reformulation of bilevel optimization to constrained optimization (Dempe & Zemkoho, 2020), and propose a novel primal-dual algorithm to solve the problem, which provably converges to an $\epsilon$-accurate KKT point. The contributions are summarized as follows.

**Algorithmic design.** We propose a simple and easy-to-implement primal-dual bilevel optimization (PDBO) algorithm, and further generalizes PDBO to its proximal version called Proximal-PDBO.

Differently from existing bilevel methods designed for handling multiple inner minima in Liu et al. (2020); Li et al. (2020); Liu et al. (2021a) that update variables $x$ and $y$ in a *nested* manner, both PDBO and Proximal-PDBO update $x$ and $y$ simultaneously as a single variable $z$ and hence admit a much simpler implementation. In addition, both algorithms do not involve any second-order information of the inner and outer functions $f$ and $g$, as opposed to many AID- and ITD-based approaches, and hence are computationally more efficient.

**Convergence rate analysis.** We provide the convergence rate analysis for PDBO and Proximal-PDBO, which serves as the first-known convergence rate guarantee for bilevel optimization with multiple inner-level minima. For PDBO, we first show that PDBO converges to an optimal solution of the associated constrained optimization problem under certain convexity-type conditions. Then, for nonconvex $f$ and convex $g$ on $y$, we show that the more sophisticated Proximal-PDBO algorithm achieves an $\epsilon$-KKT point of the reformulated constrained optimization problem for any arbitrary $\epsilon > 0$ with a sublinear convergence rate. Here, the KKT condition serves as a necessary optimality condition for the bilevel problem. Technically, the reformulated constrained problem here is more challenging than the standard constrained optimization problem studied in Boob et al. (2019); Ma et al. (2020); Gong & Liu (2021) due to the nature of bilevel optimization. Specifically, our analysis needs to deal with the bias errors arising in gradient estimations for the updates of both the primal and dual variables. Further, we establish uniform upper bound on optimal dual variables, which was taken as an assumption in the standard analysis (Boob et al., 2019).

**Empirical performance.** In two synthetic experiments with intrinsic multiple inner minima, we show that our algorithm converges to the global minimizer, whereas AID- and ITD-based methods are stuck in local minima. We further demonstrate the effectiveness and better performance of our algorithm in hyperparameter optimization.

### 1.2   Related Works

**Bilevel optimization via AID and ITD.** AID and ITD are two popular approaches to reduce the computational challenging in approximating the outer-level gradient (which is often called hypergradient in the literature). In particular, AID-based bilevel algorithms (Domke, 2012; Pedregosa, 2016; Gould et al., 2016; Liao et al., 2018; Grazzi et al., 2020b; Lorraine et al., 2020; Ji & Liang, 2021; MacKay et al., 2019) approximate the hypergraident efficiently via implicit differentiation combined with a linear system solver.

ITD-based approaches (Domke, 2012; Maclaurin et al., 2015; Franceschi et al., 2017; 2018; Shaban et al., 2019; Grazzi et al., 2020b; MacKay et al., 2019) approximate the inner-level problem using a dynamic system. For example, Franceschi et al. (2017; 2018) computed the hypergradient via reverse or forward mode in automatic differentiation. This paper proposes a novel contrained optimization based approach for bilevel optimization.

**Optimization theory for bilevel optimization.** Some works such as Franceschi et al. (2018); Shaban et al. (2019) analyzed the asymptotic convergence performance of AID- and ITD-based bilevel algorithms. Other works (Ghadimi & Wang, 2018; Rajeswaran et al., 2019; Grazzi et al., 2020a; Ji et al., 2020; 2021; Ji & Liang, 2021) provided convergence rate analysis for various AID- and ITD-based approaches and their variants in applications such as meta-learning. Recent works (Hong et al., 2020; Ji et al., 2021; Yang et al., 2021; Khanduri et al., 2021; Chen et al., 2021; Guo & Yang, 2021) developed convergence rate analysis for their proposed stochastic bilevel optimizers. This paper provides the first-known convergence rate analysis for the setting with multiple inner minima.

**Bilevel optimization with multiple inner minima.** Sabach and Shtern in Sabach & Shtern (2017) proposed a bilevel gradient sequential averaging method (BiG-SAM) for *single*-variable bilevel optimization (i.e., without variable $x$), and provided an asymptotic convergence analysis for this algorithm. For general bilevel problems, the authors in Liu et al. (2020); Li et al. (2020) used an idea similar to BiG-SAM, and proposed a gradient aggregation approach for the general bilevel problem in eq. (1) with an asymptotic convergence guarantee. Further, Liu et al. (2021a) proposed a constrained optimization method and further applied the log-barrier interior-point method for solving the constrained problem. Differently from the above studies that update the outer variable $x$ after fully updating $y$, our PDBO and Proximal-PDBO algorithms treat both $x$ and $y$ together as a single updating variable $z$. Further, we characterize the first known convergence rate guarantee for the type of bilevel problems with multiple inner minima.

We further mention that Liu et al. (2021b) proposed an initialization auxiliary algorithm for the bilevel problems with a nonconvex inner objective function.

After this work was initially posted on arXiv, several studies along this line of research came out (Jiang et al., 2023; Xiao et al., 2023; Liu et al., 2022; 2023; Kwon et al., 2023; Chen et al., 2024). Liu et al. (2022) propose a first-order bilevel optimization method by applying the dynamic barrier gradient descent algorithm Gong et al. (2021) on the value-function reformulation of the original problem. Xiao et al. (2023) also adopt a constrained optimization reformulation and propose a general alternating method for bilevel problems with lower-level objective that satisfies the PL condition. Chen et al. (2024) adopt the standard hyper-objective approach but without the typical lower-level strong convexity assumption.

## 2   Problem Formulation

We study a bilevel optimization problem given in eq. (1), which is restated below

$$\min_{x \in \mathcal{X}, y \in \mathcal{S}_x} f(x, y) \quad \text{with} \quad \mathcal{S}_x = \arg\min_{y \in \mathbb{R}^d} g(x, y),$$

where the outer- and inner-level objective functions $f(x, y)$ and $g(x, y)$ are continuously differentiable, and the supports $\mathcal{X}$ is convex and closed subsets of $\mathbb{R}^p$. For a fixed $x \in \mathcal{X}$, $\mathcal{S}_x$ is the set of all $y \in \mathbb{R}^d$ that yields the minimal value of $g(x, \cdot)$. In this paper, we consider the function $g$ that is a convex function on $y$ for any fixed $x$ (as specified in Assumption 1). The convexity of $g(x, y)$ on $y$ still allows the inner function $g(x, \cdot)$ to have multiple global minimal points, and the challenge for bilevel algorithm design due to multiple inner minima still remains. Further, the set $\mathcal{S}_x$ of minimizers is convex due to convexity of $g(x, y)$ w.r.t. $y$. We note that $g(x, y)$ and the outer function $f(x, y)$ can be nonconvex w.r.t $(x, y)$ in general or satisfy certain convexity-type conditions which we will specify for individual cases. We further take the standard gradient Lipschitz assumption on the inner and outer objective functions. The formal statements of our assumptions are presented below.

**Assumption 1.** *The objective functions $f(x, y)$ and $g(x, y)$ are gradient Lipschitz functions and $g(x, y)$ is twice differentiable. There exists $\rho_f, \rho_g \geq 0$, such that, for any $z = (x, y) \in \mathcal{X} \times \mathbb{R}^d$ and $z' = (x', y') \in \mathcal{X} \times \mathbb{R}^d$,*

*the following inequalities hold*

$$\|\nabla f(z) - \nabla f(z')\|_2 \leq \rho_f \|z - z'\|_2,$$
$$\|\nabla g(z) - \nabla g(z')\|_2 \leq \rho_g \|z - z'\|_2.$$

*Moreover, for any fixed $x \in \mathcal{X}$, we assume $g(x, y)$ is a convex function on $y$, and the following inequality holds for all $x, x' \in \mathcal{X}$ and $y, y' \in \mathbb{R}^d$*

$$g(x', y') \geq g(x, y) + \langle \nabla_x g(x, y), x' - x \rangle$$
$$+ \langle \nabla_y g(x, y), y' - y \rangle - \frac{\rho_g}{2} \|x - x'\|_2^2.$$

The last inequality in Assumption 1 is the geometric interpretation of gradient Lipschitz condition and the convexity of $g(x, y)$ on $y$ with fixed $x$. If $x = x'$, it equals the sufficient and necessary geometric characterization of the convexity of $g(x, y)$ on $y$. If $y = y'$, it captures Lipschitz smoothness.

To solve the bilevel problem in eq. (1), one challenge is that it is not easy to explicitly characterize the set $\mathcal{S}_x$ of the minimal points of $g(x, y)$. This motivates the idea to describe such a set implicitly via a constraint. A common practice is to utilize the so-called lower-level value function (LLVF) to reformulate the problem to an equivalent single-level optimization (Dempe & Zemkoho, 2020). Specifically, let $g^*(x) \coloneqq \min_{y \in \mathbb{R}^d} g(x, y)$. Clearly, the set $\mathcal{S}_x$ can be described as $\mathcal{S}_x = \{y \in \mathbb{R}^d : g(x, y) \leq g^*(x)\}$. In this way, the bilevel problem in eq. (1) can be equivalently reformulated to the following constrained optimization problem:

$$\min_{x \in \mathcal{X}, y \in \mathbb{R}^d} f(x, y) \quad \text{s.t.} \quad g(x, y) \leq g^*(x).$$

Since $g(x, y)$ is convex with respect to $y$, $g^*(x)$ in the constraint can be obtained efficiently via various convex minimization algorithms such as gradient descent.

To further simplify the notation, we let $z = (x, y) \in \mathbb{R}^{p+d}$, $\mathcal{Z} = \mathcal{X} \times \mathbb{R}^d$, $f(z) \coloneqq f(x, y)$, $g(z) \coloneqq g(x, y)$, and $g^*(z) \coloneqq g^*(x)$. As stated earlier, $\mathcal{X}$ is assumed to be a bounded and closed set. The equivalent single-level constrained optimization could be expressed as follows.

$$\min_{z \in \mathcal{Z}} f(z) \quad \text{s.t.} \quad h(z) \coloneqq g(z) - g^*(z) \leq 0. \tag{2}$$

Thus, solving the bilevel problem in eq. (1) is converted to solving an equivalent single-level optimization in eq. (2). To enable the algorithm design for the constrained optimization problem in eq. (2), we further make two standard changes to the constraint function. (i) Since the constraint is nonsmooth, i.e., $g^*(z) \coloneqq g^*(x)$ is nonsmooth in general, the design of gradient-based algorithm is not direct. We thus relax the constraint by replacing $g^*(z)$ with a smooth term

$$\tilde{g}^*(z) \coloneqq \tilde{g}^*(x) = \min_{y \in \mathbb{R}^d} \left\{ \tilde{g}(x, y) \coloneqq g(x, y) + \frac{\alpha}{2} \|y\|^2 \right\},$$

where $\alpha > 0$ is a small prescribed constant. It can be shown that for a given $x$, $\tilde{g}(x, y)$ has a unique minimal point, and the function $\tilde{g}^*(x)$ becomes differentiable with respect to $x$. Hence, the constraint becomes $g(z) - \tilde{g}^*(z) \leq 0$, which is differentiable. (ii) The constraint is not sufficiently strictly feasible, i.e., the constraint cannot be satisfied with a certain margin on every $x \in \mathcal{X}$, due to which it is difficult to design an algorithm with convergence guarantee. We hence further relax the constraint by introducing a positive small constant $\delta$ so that the constraint becomes $g(z) - \tilde{g}^*(z) - \delta \leq 0$ that admits strict feasible points such that the constraint is less than $-\delta$. Given the above two relaxations, our algorithm design will be based on the following best-structured constrained optimization problem

$$\min_{z \in \mathcal{Z}} f(z) \quad \text{s.t.} \quad \tilde{h}(z) \coloneqq g(z) - \tilde{g}^*(z) - \delta \leq 0. \tag{3}$$

We next characterize the connections of the KKT points of Equations (2) and (3) below.

**Proposition 1.** *Suppose Assumption 1 holds. If $z_{\hat{k}} = (x_{\hat{k}}, y_{\hat{k}})$ is an $\epsilon$-KKT point of the problem in eq. (3), it is also an $\tilde{\epsilon}$-KKT point of eq. (2) with $\tilde{\epsilon} = \mathcal{O}(\epsilon + \alpha + \delta)$.*

Clearly, if we set $\alpha = \mathcal{O}(\epsilon)$ and $\delta = \mathcal{O}(\epsilon)$, then we also obtain an $\epsilon$-KKT point of eq. (2).

# 3 Primal-Dual Bilevel Optimizer (PDBO)

In this section, we first propose a simple PDBO algorithm, and then show that PDBO converges under certain convexity-type conditions. We handle more general $f$ and $g$ in Section 4.

## 3.1 PDBO Algorithm

To solve the constrained optimization problem in eq. (3), we employ the primal-dual approach. The idea is to consider the following dual problem

$$\max_{\lambda \geq 0} \min_{z \in \mathcal{Z}} \mathcal{L}(z, \lambda) = f(z) + \lambda \tilde{h}(z), \tag{4}$$

where $\mathcal{L}(z, \lambda)$ is called the Lagrangian function and $\lambda$ is the dual variable. A simple approach to solving the minimax dual problem in eq. (4) is via the gradient descent and ascent method, which yields our algorithm of primal-dual bilevel optimizer (PDBO) (see Algorithm 1).

---

**Algorithm 1** Primal-Dual Bilevel Optimizer (PDBO)

---

1: **Input:** Stepsizes $\eta_t$ and $\tau_t$, $\theta_t$, output weights $\gamma_t$, initialization $z_0, \lambda_0$, and number $T$ of iterations
2: **for** $t = 0, 1, ..., T - 1$ **do**
3:     Conduct projected gradient descent using eq. (5) for $N$ times with any given $\hat{y}_0$ as initialization
4:     Update $\lambda_{t+1}$ according to eq. (6)
5:     Update $z_{t+1}$ according to eq. (7)
6: **end for**
7: **Output:** $\bar{z} = \frac{1}{\Gamma_T} \sum_{t=0}^{T-1} \gamma_t z_{t+1}$, with $\Gamma_T = \sum_{t=0}^{T-1} \gamma_t$

---

More specifically, updates of the dual variable $\lambda$ are via the gradients of Lagrangian w.r.t. $\lambda$ given by $\nabla_\lambda \mathcal{L}(z, \lambda) = \tilde{h}(z)$, and updates of the primal variable $z$ are via the gradients of Lagrangian w.r.t. $z$ given by $\nabla_z \mathcal{L}(z, \lambda) = \nabla f(z) + \lambda \nabla \tilde{h}(z)$. Here, the differentiability of $\tilde{h}(z)$ benefits from the constraint smoothing. In particular, suppose Assumption 1 holds. Then, it can be easily shown that $\tilde{h}(z)$ is differentiable and $\nabla_x \tilde{h}(x, y) = \nabla_x g(x, y) - \nabla_x g(x, \tilde{y}^*(x))$, where $\tilde{y}^*(x) = \arg\min_{y \in \mathbb{R}^d} \tilde{g}(x, y)$ is the unique minimal point of $\tilde{g}(x, y)$. Together with the fact that $\nabla_y \tilde{h}(z) = \nabla_y g(x, y)$, we have $\nabla \tilde{h}(z) = [\nabla_x g(x, y) - \nabla_x g(x, \tilde{y}^*(x)); \nabla_y g(x, y)]$. Since $\tilde{y}^*(x)$ is the minimal point of the inner problem: $\min_{y \in \mathbb{R}^d} g(x_t, y) + \frac{\alpha}{2} \|y\|_2^2$, we conduct $N$ steps of projected gradient descent

$$\hat{y}_{n+1} = \hat{y}_n - \frac{2}{\rho_g + 2\alpha} \left( \nabla_y g(x_t, \hat{y}_n) + \alpha \hat{y}_n \right), \tag{5}$$

and take $\hat{y}_N$ as an estimate of $\tilde{y}^*(x_t)$. Since the inner function $\tilde{g}(x, y)$ is $\alpha$-strongly convex w.r.t. $y$, updates in eq. (5) converge exponentially fast to $\tilde{y}^*(x_t)$ w.r.t. $N$. Hence, with only a few steps, we can obtain a good estimate. With the output $\hat{y}_N$ of eq. (5) as the estimate of $\tilde{y}^*(x_t)$, we conduct the accelerated projected gradient ascent and projected gradient descent as follows:

$$\lambda_{t+1} = \Pi_\Lambda \left( \lambda_t + \frac{1}{\tau_t} \left( (1 + \theta_t) \hat{h}(z_t) - \theta_t \hat{h}(z_{t-1}) \right) \right), \tag{6}$$

$$z_{t+1} = \Pi_{\mathcal{Z}} \left( z_t - \frac{1}{\eta_t} \left( \nabla f(z_t) + \lambda_{t+1} \hat{\nabla} \tilde{h}(z_t) \right) \right), \tag{7}$$

where $\frac{1}{\tau_t}, \frac{1}{\eta_t}$ are the stepsizes, $\theta_t$ is the acceleration weight, $\Lambda = [0, B]$, with $B > 0$ being a prescribed constant, $\hat{h}(z_t) = g(x_t, y_t) - \tilde{g}(x_t, \hat{y}_N)$, and

$$\hat{\nabla} \tilde{h}(z_t) = \nabla g(z_t) - \begin{pmatrix} \nabla_x g(x_t, \hat{y}_N) \\ \mathbf{0}_d \end{pmatrix}$$

where $\mathbf{0}_d$ is the all zero vector in $\mathbb{R}^d$.

## 3.2 Convergence Rate of PDBO

As formulated in Section 2, the problem in eq. (3) in general can be a nonconvex objective and nonconvex constrained optimization under Assumption 1. For such a problem, the gradient descent with respect to $z$

can guarantee only the convergence $\|\nabla_z \mathcal{L}(z, \lambda)\|_2 \to 0$. Here, the updates of $\lambda$ change only the weight that the gradient of the constraint contributes to the gradient of the Lagrangian, which does not necessarily imply the convergence of the function value of the constraint. Thus, we anticipate PDBO to converge under further geometric requirements as stated below.

**Assumption 2.** *The objective function $f(z)$ is a $\mu$-strongly convex function with respect to $z$, and the constrained function $\tilde{h}(z)$ is a convex function on $z$.*

Under Assumption 2, the global optimal point exists and is unique. Let such a point be $z^*$. We provides the convergence result with respect to such a point below.

**Theorem 1.** *Suppose Assumptions 1 and 2 hold. Consider Algorithm 1. Let $B > 0$ be some large enough constant, $\gamma_t = \mathcal{O}(t)$, $\eta_t = \mathcal{O}(t)$, $\tau_t = \mathcal{O}(\frac{1}{t})$ and $\theta_t = \gamma_{t+1}/\gamma_t$, where the exact expressions can be found in the appendix. Then, the output $\bar{z}$ of PDBO converges to $z^*$, which satisfies*
$$\max\{f(\bar{z}) - f(z^*), [\tilde{h}(\bar{z})]_+, \|\bar{z} - z^*\|_2^2\} \leq \mathcal{O}\left(\tfrac{1}{T^2}\right) + \mathcal{O}\left(e^{-N}\right)$$
*with $[x]_+ = \max\{x, 0\}$.*

Theorem 1 indicates that all of the optimality gap $f(\bar{z}) - f(z^*)$, the constraint violation $[\tilde{h}(\bar{z})]_+$, and the squared distance $\|\bar{z} - z^*\|_2^2$ between the output and the optimal point converge sublinearly as the number of iterations enlarges. In particular, the first term of the bound captures the accumulated gap of the outer function values among iterations, and the second term is due to the biased estimation of $\tilde{y}^*(x_t)$ in each iteration.

**Corollary 1.** *By setting $T = \mathcal{O}(\frac{1}{\sqrt{\epsilon}})$ and $N = \mathcal{O}(\log(\frac{1}{\epsilon}))$, Theorem 1 ensures that $\bar{z}$ is an $\epsilon$-optimal point of the constrained problem in eq. (3), i.e. the optimality gap $f(\bar{z}) - f(z^*)$, constraint violation $[\tilde{h}(\bar{z})]_+$, and squared distance $\|\bar{z} - z^*\|_2^2$ are all upper-bounded by $\epsilon$. Moreover, the total complexity of gradient accesses is $TN = \tilde{O}(\frac{1}{\sqrt{\epsilon}})$.*

We remark that our proof here is more challenging than the generic constrained optimization (Boob et al., 2019; Ma et al., 2020) due to the nature of the bilevel optimization. Specifically, the constraint function here includes the minimal value $y^*(x_t) \coloneqq \arg\min_{y \in \mathbb{R}^d} \tilde{g}(x_t, y)$ of the inner function, where both its value and the minimal point will be estimated during the execution of algorithm, which will cause the gradients of both primal and dual variables to have bias errors. Our analysis will need to deal with such bias errors and characterize their impact on the convergence.

We further remark that although PDBO has guaranteed convergence under convexity-type conditions, it can still converge fast under more general problems when $f$ and $g$ are nonconvex as we demonstrate in our experiments in Section 5 and in appendix. However, formal theoretical treatment of nonconvex problems will require more sophisticated design as we present in the next section.

## 4 Proximal-PDBO Algorithm

### 4.1 Algorithm Design

In the previous section, we introduce PDBO and provide its convergence rate under convexity-type conditions. In order to solve the constrained optimization problem eq. (3) in the general setting, we will adopt the proximal method (Boob et al., 2019; Ma et al., 2020). The general idea is to iteratively solve a series of sub-problems, constructed by regularizing the objective and constrained functions into strongly convex functions. In this way, the algorithm is expected to converge to a stochastic $\epsilon$-KKT point (see Definition 1 in Section 4.2) of the primal problem in eq. (3).

By applying the proximal method, we obtain the Proximal-PDBO algorithm (see Algorithm 2) for solving the bilevel optimization problems formulated in eq. (3). At each iteration, the algorithm first constructs two proximal functions corresponding to the objective $f(z)$ and constraint $\tilde{h}(z)$ via regularizers. Since $f(z)$ is $\rho_f$-gradient Lipschitz as assumed in Assumption 1, the constructed function $f_k(z)$ is strongly convex with $\mu = \rho_f$. For the new constraint $\tilde{h}_k(z)$, we next show that it is a convex function with large enough regularization coefficient $\rho$.

---

**Algorithm 2** Proximal-PDBO Algorithm

---

1: **Input:** Stepsizes $\eta_t$ and $\tau_t$, $\theta_t$, output weights $\gamma_t$ and iteration numbers $K$, and $T$
2: Set $\tilde{z}_0$ be any point inside $\mathcal{Z}$
3: **for** $k = 1, ..., K$ **do**
4:      Set the sub-problem

$$\min_{z \in \mathcal{Z}} \, f_k(z) \coloneqq f(z) + \rho_f \|z - \tilde{z}_{k-1}\|_2^2,$$

$$\text{s.t. } \tilde{h}_k(z) \coloneqq \tilde{h}(z) + \rho \|x - \tilde{x}_{k-1}\|_2^2 \le 0. \tag{$P_k$}$$

5:      Internalize $z_0 = z_{-1} = \tilde{z}_{k-1}$ and $\lambda_0 = \lambda_{-1} = 0$
6:      **for** $t = 0, 1, ..., T-1$ **do**
7:          Conduct updates of $\hat{y}_t$ using eq. (5) for $N$ times with any initial point $\hat{y}_0$ to estimate $\tilde{y}^*(x_t)$
8:          Update $\lambda_{t+1}$ according to eq. (8)
9:          Update $z_{t+1}$ according to eq. (9)
10:      **end for**
11:      Set $\tilde{z}_k = \frac{1}{\Gamma_T} \sum_{t=0}^{T-1} \gamma_t z_{t+1}$, with $\Gamma_T = \sum_{t=0}^{T-1} \gamma_t$.
12: **end for**
13: Randomly pick $\hat{k}$ from $\{1, \ldots, K\}$
14: **Output:** $\tilde{z}_{\hat{k}}$

---

**Lemma 1.** *Suppose that Assumption 1 holds. Let* $\rho = \frac{2\alpha\rho_g + \rho_g^2}{2\alpha}$, *then* $\tilde{h}_k(z)$ *is a convex function.*

The above lemma and the $\rho_f$-strong convexity of $f_k(z)$ ensure that Assumption 2 holds with $f_k(z)$ and $h_k(z)$. Then lines 5-10 adopt the PDBO as a subroutine to solve the subproblem ($P_k$), which is shown to be effective in Theorem 1. Similarly to what we have done in Section 3, we estimate $\tilde{h}_k(z_t)$ and $\nabla\tilde{h}_k(z_t)$ using $\hat{y}_N$ as $\hat{h}_k(z_t) = g(x_t, y_t) - \tilde{g}(x_t, \hat{y}_N) - \delta + \rho_h\|x_t - \tilde{x}_k\|_2^2$, and $\hat{\nabla}\tilde{h}_k(z_t) = \nabla g(z_t) - (\nabla_x\tilde{g}(x_t, \hat{y}_N); \mathbf{0}_d) + 2\rho(x_t - \tilde{x}_k; \mathbf{0}_d)$.

The gradient of the Lagrangian is immediately obtained through $\hat{\nabla}_\lambda \mathcal{L}_k(z_t, \lambda_t) = \hat{h}_k(z_t)$ and $\hat{\nabla}_z \mathcal{L}_k(z_t, \lambda_{t+1}) = \nabla f_k(z_t) + \lambda_{t+1}\hat{\nabla}\tilde{h}_k(z_t)$. Finally, the main step of updating dual and primal variables in eqs. (6) and (7) are adjusted here as follows:

$$\lambda_{t+1} = \Pi_\Lambda\left(\lambda_t + \tfrac{1}{\tau_t}\left((1+\theta_t)\hat{h}_k(z_t) - \theta_t\hat{h}_k(z_{t-1})\right)\right), \tag{8}$$

$$z_{t+1} = \Pi_\mathcal{Z}\left(z_t - \tfrac{1}{\eta_t}\hat{\nabla}_z\mathcal{L}_k(z_t, \lambda_{t+1})\right). \tag{9}$$

## 4.2 Convergence Rate of Proximal-PDBO

We first introduce the following first-order necessary condition of optimality for the nonconvex optimization problem with nonconvex constraint in eq. (3) (Boob et al., 2019; Ma et al., 2020).

**Definition 1** ((Stochastic) $\epsilon$-KKT point). *Consider the constrained optimization problem in eq. (3). A point* $\hat{z} \in \mathcal{Z}$ *is an* $\epsilon$-KKT *point iff., there exist* $z \in \mathcal{Z}$ *and* $\lambda \ge 0$ *such that* $\tilde{h}(z) \le 0$, $\|z - \hat{z}\|_2^2 \le \epsilon$, $|\lambda\tilde{h}(z)| \le \epsilon$, *and* $\text{dist}\left(\nabla f(z) + \nabla\tilde{h}(z), -\mathcal{N}(z; \mathcal{Z})\right) \le \epsilon$, *where* $\mathcal{N}(z; \mathcal{Z})$ *is the normal cone to* $\mathcal{Z}$ *at* $z$, *and the distance between a vector* $v$ *and a set* $\mathcal{V}$ *is* $\text{dist}(v, \mathcal{V}) \coloneqq \inf\{\|v - v'\|_2 : v' \in \mathcal{V}\}$. *For random* $\hat{z} \in \mathcal{Z}$, *it is a stochastic* $\epsilon$-KKT *point if there exist* $z \in \mathcal{Z}$ *and* $\lambda \ge 0$ *such that the same requirements of* $\epsilon$-KKT *hold in expectation.*

We will take the $\epsilon$-KKT condition as our convergence metric. It has been shown that the above KKT condition serves as the first-order necessary condition for the optimality guarantee for nonconvex constrained optimization under the MFCQ condition(Mangasarian & Fromovitz, 1967).

Next, we establish the convergence guarantee for Proximal-PDBO, which does not follow directly from that for standard constrained nonconvex optimization (Boob et al., 2019; Ma et al., 2020) due to the special challenges arising in bilevel problem formulations. Our main development lies in showing that the optimal dual variables for all subproblems visited during the algorithm iterations are uniformly bounded. Then the convergence of the Proximal-PDBO follows from the convergence of each subproblem (which we establish in Theorem 1) and the uniform bound of optimal dual variables. The details of the proof could be found in the appendix.

**Theorem 2.** *Suppose Assumption 1 holds. Consider Algorithm 2. Let the hyperparameters* $B > 0$ *be a large enough constant,* $\gamma_t = \mathcal{O}(t)$, $\eta_t = \mathcal{O}(t)$, $\tau_t = \mathcal{O}(\frac{1}{t})$ *and* $\theta_t = \gamma_{t+1}/\gamma_t$. *Then, the output* $\tilde{z}_{\hat{k}}$ *of*

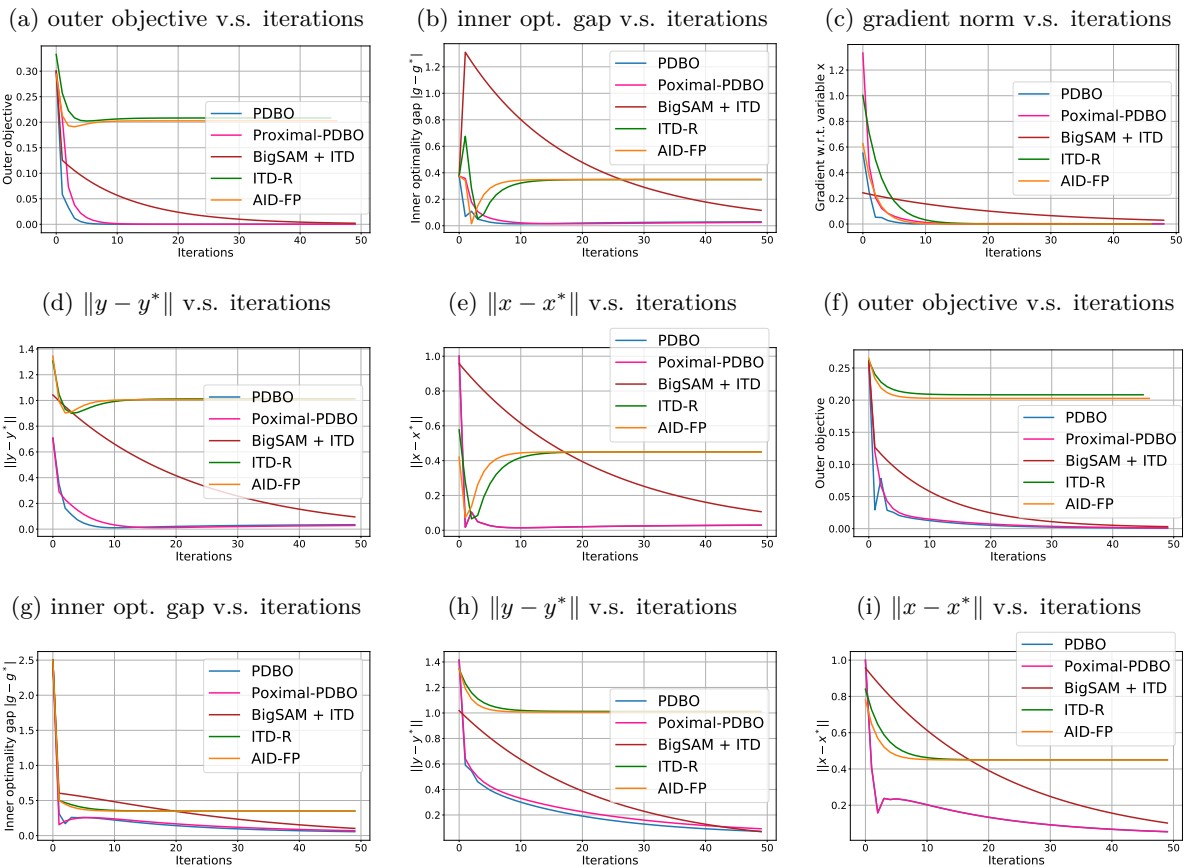

Figure 1: Evaluation of algorithms using different optimality metrics. First row and first two plots in 2nd row: $(x_0, y_0) = (2, (0.5, 0.5))$. Last plot in 2nd and 3rd row: $(x_0, y_0) = (0, (2, 2))$. PDBO is always initialized with $\lambda = 2$.

*Algorithm 2 with a randomly chosen index $\hat{k}$ is a stochastic $\epsilon$-KKT point of eq. (3), where $\epsilon$ is given by $\epsilon = \mathcal{O}\left(\frac{1}{K}\right) + \mathcal{O}\left(\frac{1}{T^2}\right) + \mathcal{O}\left(e^{-N}\right)$.*

Theorem 2 characterizes the convergence of Proximal-PDBO. In particular, there are three sources of the convergence error: (a) the inaccurate initialization of proximal center $\tilde{z}_0$ captured by $\mathcal{O}(\frac{1}{K})$, (b) the distance between $z_0$ and the optimal point of each subproblem $(\text{P}_k)$ upper-bounded by $\mathcal{O}(\frac{1}{T^2})$, and (c) the inaccurate estimation of $\tilde{y}^*(x_t)$ in the updates captured by $\mathcal{O}(e^{-N})$.

**Corollary 2.** *Theorem 2 indicates that for any prescribed accuracy level $\epsilon > 0$, by setting $K = \mathcal{O}(\frac{1}{\epsilon})$, $T = \mathcal{O}(\frac{1}{\sqrt{\epsilon}})$ and $N = \mathcal{O}(\log(\frac{1}{\epsilon}))$, we obtain an $\epsilon$-KKT point in expectation. The total computation of gradients is given by $KTN = \tilde{\mathcal{O}}(\frac{1}{\epsilon^{3/2}})$.*

We further note that Theorems 1 and 2 are the first known finite-time convergence rate characterization for bilevel optimization problems with multiple inner minimal points.

## 5 Experiments

In this section, we first consider the numerical verification for our algorithm over two synthetic problems, where one of them is moved to the appendix due to the page limit, and then apply it to hyperparameter optimization.

### 5.1 Numerical Verification

Consider the following bilevel optimization problem:

$$\min_{x \in \mathbb{R}} \ f(x, y) \coloneqq \frac{1}{2}\|(1, x)^\top - y\|^2$$
$$\text{s.t. } \ y \in \arg\min_{y \in \mathbb{R}^2} \ g(x, y) \coloneqq \frac{1}{2}y_1^2 - xy_1, \tag{10}$$

where $y$ is a vector in $\mathbb{R}^2$ and $x \in [-100, 100]$ is a scalar. It is not hard to analytically derive that the optimal solution of the problem in eq. (10) is $(x^*, y^*) = (1, (1, 1))$, which corresponds to the optimal objective values of $f^* = 0$ and $g^* = -\frac{1}{2}$. For a given value of $x$, the lower-level problem admits a unique minimal value $g^*(x) = -\frac{1}{2}x$, which is attained at all points $y = (x, a)^\top$ with $a \in \mathbb{R}$. Hence, the problem in eq. (10) violates the requirement of the existence of a **single** minimizer for the inner-problem, which is a strict requirement for most existing bilevel optimization methods, but still fall into our theoretical framework that allows multiple inner minimizers. In fact, it can be analytically shown that standard AID and ITD approaches cannot solve the problem in eq. (10), which makes it both interesting and challenging.

We compare our algorithms **(Proximal-)PDBO** with the following representative methods for bilevel optimization:

- **BigSAM + ITD** (Liu et al., 2020; Li et al., 2020): uses sequential averaging to solve the inner problem and applies reverse mode automatic differentiation to compute hypergradient. This method is also designed to solve bilevel problems with multiple inner minima.

- **AID-FP** (Grazzi et al., 2020b): an approximate implicit differentiation approach with Hessian inversion using fixed point method. This method is guaranteed to converge when the lower-level problem admits a unique minimizer.

- **ITD-R** (Franceschi et al., 2017): the standard iterative differentiation method for bilevel optimization, which differentiates through the unrolled inner gradient descent steps. We use its reverse mode implementation. This method is also guaranteed to converge when the lower level problem admits a unique minimizer.

For our **PDBO**, we set the learning rates $\tau_t$, $\eta_t$ to be constants 0.1, 0.2 respectively, and $\theta_t = 0$. For our **Proximal-PDBO**, we set the $\tau_t$, $\eta_t$ and $\theta_t$ to be the same as **PDBO**. Moreover, we specify $T = 50$. For all compared methods, we fix the inner and outer learning rates to respectively 0.5 and 0.2. We use $N = 5$ gradient descent steps to estimate the minimimal value of the smoothed inner-objective and use the same number of iterations for all compared methods.

Figure 1 shows several evaluation metrics for the algorithms under comparison over different initialization points. It can be seen that our two algorithms **(Proximal-)PDBO** reach the optimal solution at the fastest rate. Also as analytically proved in Liu et al. (2020), several plots show that, with different initialization points, the classical AID and ITD methods cannot converge to the global optimal solution of the problem in eq. (10). In particular, algorithms **AID-FP** and **ITD-R** are both stuck in a bad local minima of the problem (fig. 1 (c), (d), (e), (h), and (i)). This is essentially due to the very restrictive unique minimizer assumption of these methods. When this fundamental requirement is not met, such approaches are not equipped with mechanisms to select among the multiple minimizers. Instead, our algorithm, which solves a constrained optimization problem, leverages the constraint set to guide the optimization process. In the appendix, we have provide evidence that our **PDBO** do converge to a KKT point of the reformulated problem in eq. (3) for the problem eq. (10).

**Further Experiments.** Besides the example in eq. (10), we conduct another experiment (presented in the appendix) to demonstrate that in practice **PDBO** can converge well for a broader class of functions even when $g(x, y)$ is not convex on $y$. The results show that **PDBO** outperforms all state-of-the-art methods with a large margin.

### 5.2 Hyperparameter Optimization

The goal of hyperparameter optimization (HO) is to search for the set of hyperparameters that yield the optimal value of some model selection criterion (e.g., loss on unseen data). HO can be naturally expressed as

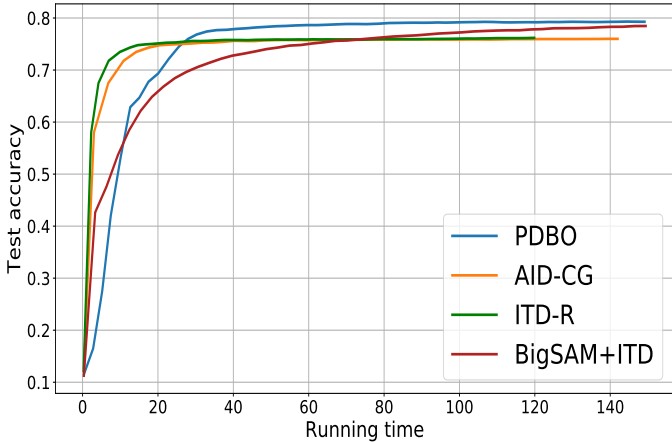

Figure 2: Accuracy on test data on the 20 Newsgroup dataset

| Algo. | PDBO | AID-CG | ITD-R | BigSAM+ITD |
|-------|------|--------|-------|------------|
| Acc. | **79.26** | 76.20 | 76.20 | 78.92 |

Table 1: Test accuracy on the 20 Newsgroup dataset.

a bilevel optimization problem, in which at the inner level one searches for the model parameters that achieve the lowest training loss for given hyperparameters. At the outer level, one optimizes the hyperparameters over a validation dataset. The problem can be mathematically formulated as follows

$$\min_{\lambda \in \Lambda, w \in \mathcal{W}(\lambda)} \mathcal{L}_{\mathrm{val}}(\lambda, w) := \frac{1}{|\mathcal{D}_{\mathrm{val}}|} \sum_{\xi \in \mathcal{D}_{\mathrm{val}}} \mathcal{L}(\lambda, w; \xi),$$

$$\text{with } \mathcal{W}(\lambda) := \arg\min_{w} \mathcal{L}_{\mathrm{tr}}(\lambda, w),$$

where $\mathcal{L}_{\mathrm{tr}}(\lambda, w) := \frac{1}{|\mathcal{D}_{\mathrm{tr}}|} \sum_{\zeta \in \mathcal{D}_{\mathrm{tr}}} (\mathcal{L}(\lambda, w; \zeta) + \mathcal{R}(\lambda, w))$, $\mathcal{L}$ is a loss function, $\mathcal{R}(w, \lambda)$ is a regularizer, and $\mathcal{D}_{\mathrm{tr}}$ and $\mathcal{D}_{\mathrm{val}}$ are respectively training and validation data.

Following Franceschi et al. (2017); Grazzi et al. (2020a), we perform classification on the 20 Newsgroup dataset, where the classifier is modeled by an affine transformation and the cost function $\mathcal{L}$ is the cross-entrpy loss. We set one $\ell_2$-regularization hyperparameter for each weight in $w$, so that $\lambda$ and $w$ have the same size, and bound the search space such that $\|\Lambda\|_\infty \leq 1000$. For our algorithm **PDBO**, we optimize the parameters and hyperparameters using gradient descent with a fixed learning rate of $\eta_t^{-1} = 100$. We set the learning rate for the dual variable to be $\tau_t^{-1} = 0.001$. We use $N = 5$ gradient descent steps to estimate the minimal value of the smoothed inner-objective. For **BigSAM+ITD**, we set the averaging parameter to 0.5 and fix the inner and outer learning rates to be 100. For **AID-CG** and **ITD-R**, we use the suggested parameters in their implementations accompanying the paper (Grazzi et al., 2020a).

The evaluations of the algorithms under comparison on a hold-out test dataset is shown in Figure 2 and Table 1. It can be seen that our algorithm **PDBO** significantly improves over AID/ITD methods, and slightly outperforms **BigSAM+ITD** method with a much faster speed.

## 5.3 Hyper-representation Learning

The hyper-representation (HR) problem (Franceschi et al., 2018; Grazzi et al., 2020a) searches for a regression (or classification) model following a two-phased optimization process. The inner-level identifies the optimal linear regressor/classifier parameters $w$, and the outer level solves for the optimal embedding model (i.e., representation) parameters $\Lambda$. Mathematically, the problem can be modeled by the following bilevel

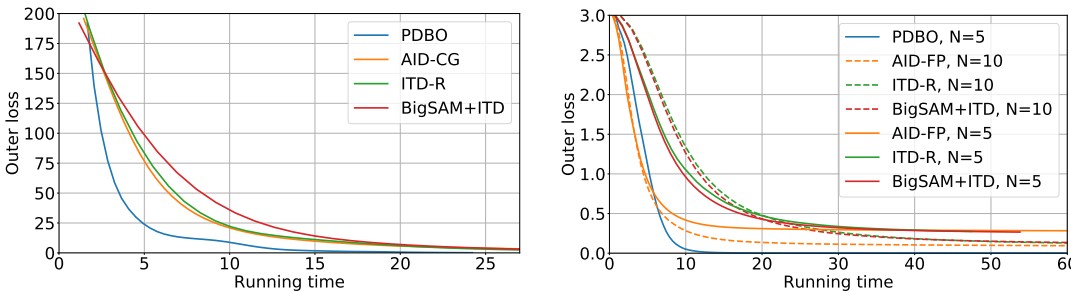

Figure 3: Hyper representation with linear embedding model. **Left plot:** Regression problem on synthetic dataset with representations dimension set to 1024. **Right plot:** Classification on 20 Newsgroup dataset with representations dimension set to 100. Our method PDBO significantly outperforms the other methods which involve second-order computations.

optimization problem:

$$\min_{\Lambda \in \mathcal{X}} f(\Lambda) = \mathcal{L}\left(h_\Lambda(X_1)w^*, Y_1\right)$$

$$\text{s.t. } w^* = \operatorname*{argmin}_{w \in \mathbb{R}^d} \mathcal{L}(h_\Lambda(X_2)w, Y_2) + \frac{\gamma}{2}\|w\|^2 \tag{11}$$

where $X_2 \in \mathbb{R}^{n_2 \times m}$ and $X_1 \in \mathbb{R}^{n_1 \times m}$ are matrices of training and validation data, $Y_2 \in \mathbb{R}^{n_2}$, $Y_1 \in \mathbb{R}^{n_1}$ are the corresponding response vectors, and $\Lambda$ corresponds to the parameters of the linear embedding model $h_\Lambda$. $\mathcal{X}$ is a subset of $\mathbb{R}^{m \times d}$ such that $\|\mathcal{X}\| \leq 10$. We consider two different settings: (a) regression on synthetic dataset in which the loss function $\mathcal{L}(\cdot, \cdot)$ corresponds to the squared $\ell_2$ norm and (b) classification on the 20 Newsgroup dataset where $\mathcal{L}(\cdot, \cdot)$ is the standard cross-entropy loss function. For the regression setting, we generate the data matrices $X_1, X_2$ and labels $Y_1, Y_1$ following the same process in Grazzi et al. (2020a). We set the dimension of the generated data as $m = 512$ and the representation dimension as $d = 1024$, so that the outer variable $\Lambda$ has the dimension $p = 512 \times 1024$ and inner variable $w$ has the dimension $d = 1024$. For the classification setting, we remove the news headers in the dataset in order to reduce the dimension of the feature vectors and pre-process the data so as to have feature vectors of dimension $m = 99238$. The representation dimension is set to $d = 100$, corresponding to the outer variable $\Lambda$ of dimension $p = 99238 \times 100$ and the inner variable $w$ of dimension $d = 100 \times 20$ for 20-ways classification.

Figure 5 shows the comparison among the different algorithms. As depicted, in both settings, our method PDBO achieves much faster convergence speed compared to all the other baselines which involve costly second-order computations. Further, the bottom plot in Figure 5 also shows that PDBO can find a good solution (i.e., the point that the algorithm finally converges to) even when a small number $N$ of inner gradient steps is employed. All the other methods require higher values of $N$ to find good solutions, which are still worse compared to the one obtained by PDBO.

## 6 Conclusion

In this paper, we investigated a bilevel optimization problem where the inner-level function has multiple minima. Based on the reformulation of such a problem as an constrained optimization, we designed two algorithms PDBO and Proximal-PDBO using primal-dual gradient descent and ascent. Specifically, PDBO features a simple design and implementation, and Proximal-PDBO features a strong convergence guarantee that we can establish. We further conducted experiments to demonstrate the desirable performance of our algorithm. As future work, it is interesting to study bilevel problems with a nonconvex inner-problem, which can have multiple inner minima. While our current design can serve as a starting point, finding a good characterization of the set of inner minima can be challenging.

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

# Supplementary Materials

## A    Comparison with Previous Bilevel Optimization Methods

We provide the comparison of our work with the state-of-the-art (SOTA) bilevel optimization methods that contain multiple minimal points in Table 2.

| | Assumptions on $f$ | Assumptions on $g$ | Convergence Results |
|---|---|---|---|
| Li et. al.Li et al. (2020) | gradient Lipschitz | Lipshitz continuous strongly convex on $y$ | asymptotic |
| Liu et. al.Liu et al. (2020) | gradient Lipschitz strongly convex on $y$ | gradient Lipschitz convex on $y$ | asymptotic |
| Liu et. al. Liu et al. (2021a) | continuously differentiable | continuously differentiable | N.A. |
| Liu et al. (2022) | gradient Lipschitz | gradient Lipschitz local PL inequality | non-asymptotic |
| Xiao et al. (2023) | gradient Lipschitz | gradient Lipschitz PL inequality | non-asymptotic |
| Our work | gradient Lipschitz | gradient Lipschitz convex on $y$ | non-asymptotic |

Table 2: Comparison between the SOTA bilevel optimization methods and our work

## B    Optimization Paths of PDBO

In Figure 4, we plot the optimization paths of **PDBO** when solving the problem in eq. (10). The two plots show that the optimization terminates with a strictly positive dual variable and that the constraint is satisfied with equality. Thus, the slackness condition is satisfied. This also confirms that our algorithm **PDBO** did converge to a KKT point of the reformulated problem.

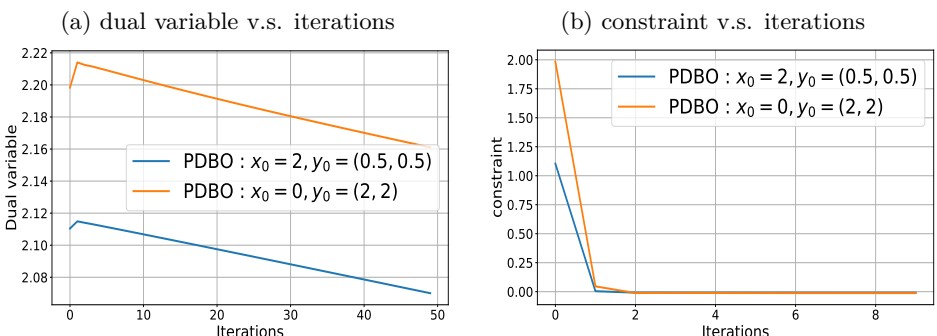

Figure 4: Optimization path of dual variable and constraint values for different initializations.

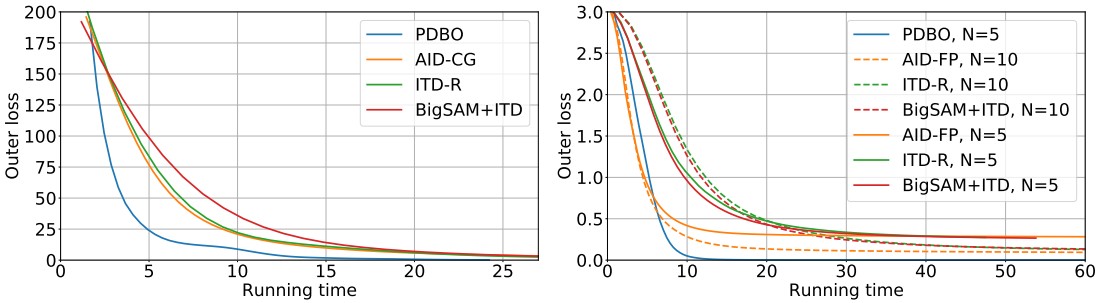

Figure 5: Hyper representation with linear embedding model. **Top plot:** Regression problem on synthetic dataset: the representations dimension is set to 1024. **Bottom plot:** Classification problem on 20 Newsgroup dataset: representations dimension is set to 100. Our method PDBO significantly outperforms the other methods which involve second-order computations. The bottom plot also shows that PDBO can find a good solution even when a small number $N$ of inner gradient steps is employer whereas the other methods require higher values of N to find solutions that are still worse compared to the one obtained by PDBO.

## C   Additional Experiment on Hyper-representation

The hyper-representation (HR) problem Franceschi et al. (2018); Grazzi et al. (2020a) searches for a regression (or classification) model following a two-phased optimization process. The inner-level identifies the optimal linear regressor/classifier parameters $w$, and the outer level solves for the optimal embedding model (i.e., representation) parameters $\Lambda$. Mathematically, the problem can be modeled by the following bilevel optimization problem:

$$
\min_{\Lambda \in \mathbb{R}^{m \times d}} f(\Lambda) = \mathcal{L}\left(h_\Lambda(X_1)w^*, Y_1\right)
$$

$$
\text{s.t.}\;\; w^* = \operatorname*{argmin}_{w \in \mathbb{R}^d} \mathcal{L}(h_\Lambda(X_2)w, Y_2) + \frac{\gamma}{2}\|w\|^2 \tag{12}
$$

where $X_2 \in \mathbb{R}^{n_2 \times m}$ and $X_1 \in \mathbb{R}^{n_1 \times m}$ are matrices of training and validation data, $Y_2 \in \mathbb{R}^{n_2}$, $Y_1 \in \mathbb{R}^{n_1}$ are the corresponding response vectors, and $\Lambda$ corresponds to the parameters of the linear embedding model $h_\Lambda$. We consider two different settings: (a) regression on synthetic dataset in which the loss function $\mathcal{L}(\cdot, \cdot)$ corresponds to the squared $\ell_2$ norm and (b) classification on the 20 Newsgroup dataset where $\mathcal{L}(\cdot, \cdot)$ is the standard cross-entropy loss function. For the regression setting, we generate the data matrices $X_1, X_2$ and labels $Y_1, Y_1$ following the same process in Grazzi et al. (2020a). We set the dimension of the generated data as $m = 512$ and the representation dimension as $d = 1024$, so that the outer variable $\Lambda$ has the dimension $p = 512 \times 1024$ and inner variable $w$ has the dimension $d = 1024$. For the classification setting, we remove the news headers in the dataset in order to reduce the dimension of the feature vectors and pre-process the data so as to have feature vectors of dimension $m = 99238$. The representation dimension is set to $d = 100$, corresponding to the outer variable $\Lambda$ of dimension $p = 99238 \times 100$ and the inner variable $w$ of dimension $d = 100 \times 20$ for 20-ways classification.

Figure 5 shows the comparison among the different algorithms. As depicted, in both settings, our method PDBO achieves much faster convergence speed compared to all the other baselines which involve costly second-order computations. Further, the bottom plot in Figure 5 also shows that PDBO can find a good solution (i.e., the point that the algorithm finally converges to) even when a small number $N$ of inner gradient steps is employed. All the other methods require the same or higher values of $N$ to find good solutions, which are still worse compared to the one obtained by PDBO.

# D    Additional Experiment with Multiple Minima

In this section, we demonstrate that **(Proximal-)PDBO** is applicable to a more general class of bilevel problems in practice, where the inner problem is not necessarily convex on $y$ (as we require in the theoretical analysis), but still have multiple minimal points.

Consider the following bilevel optimization problem

$$\min_{x \in \mathcal{C}, y \in \mathcal{S}_x} \|x - a\|^2 + \|y - a\|^2, \quad \text{where} \quad \mathcal{S}_x := \arg\min_{y \in \mathcal{C}} \sin(x + y), \tag{13}$$

where $\mathcal{C} = [-10, 10]$, and $a$ is a constant (we set $a = 0$ in our experiment).

Clearly, the problem in eq. (13) does not degenerate to single inner minimum bilevel optimization due to the sinusoid in the lower function. Such a problem is harder than the one in eq. (10), and it violates the assumption that $g(x, y)$ is convex on $y$.

In this experiment, we compare **PDBO**, **Proximal-PDBO**, **BigSAM+ITD**, **ITD-R** and **AID-FP**. We initialize all methods with $(x, y) = (3, 3)$ and the hyperparameters are set to be the same as what we did in Section 5. The results are provided in Figure 6. The plots show that **ITD-R** and **AID-FP** methods converge to a bad stationary point and could not find good solutions. The methods **PDBO**, **Proximal-PDBO**, and **BigSAM+ITD** converge to better points but our algorithms **PDBO** and **Proximal-PDBO** significantly outperform **BigSAM+ITD**.

(a) outer optimality gap    (b) outer convergence error    (c) inner convergence error

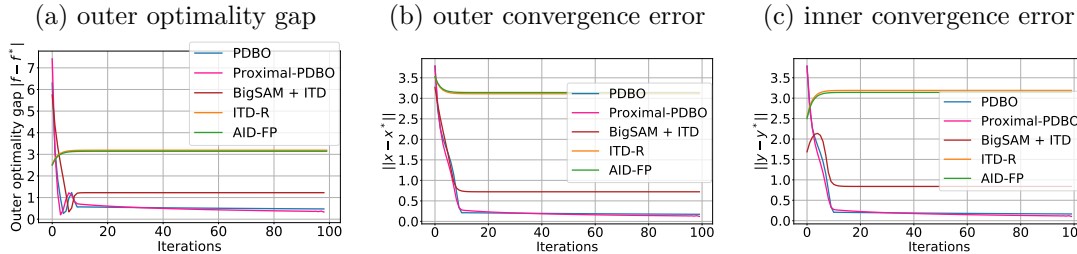

Figure 6: Comparison of different algorithms

# E    Calculation of $\nabla \tilde{h}(z)$ in Section 3

For completeness, we provide the steps for obtaining the form of $\nabla \tilde{h}(z)$ in Section 2. For the ease of reading, we restate the result here. Suppose that Assumption 1 holds. Then the gradients $\nabla_x \tilde{h}(x, y)$ and $\nabla_y \tilde{h}(x, y)$ of function $\tilde{h}(x, y)$ take the following forms:

$$\nabla_x \tilde{h}(x, y) = \nabla_x g(x, y) - \nabla_x g(x, \tilde{y}^*(x)), \tag{14}$$

$$\nabla_y \tilde{h}(x, y) = \nabla_y g(x, y). \tag{15}$$

*Proof.* First, eq. (15) follows immediately. Hence, we prove only eq. (14). Recall the definition of $\tilde{h}(x, y)$:

$$\tilde{h}(x, y) = g(x, y) - \tilde{g}^*(x) - \delta,$$

where $\tilde{g}^*(x) = \tilde{g}(x, \tilde{y}^*(x))$ with $\tilde{y}^*(x) = \arg\min_y \tilde{g}(x, y) = g(x, y) + \frac{\alpha}{2}\|y\|^2$. Using the chain rule to compute the gradient with respect to $x$ of function $\tilde{h}(x, y) = g(x, y) - g(x, \tilde{y}^*(x)) - \frac{\alpha}{2}\|\tilde{y}^*(x)\|^2 - \delta$ yields:

$$\nabla_x \tilde{h}(x, y) = \nabla_x g(x, y) - \left[\nabla_x g(x, \tilde{y}^*(x)) + \frac{\partial \tilde{y}^*(x)}{\partial x} \nabla_y g(x, \tilde{y}^*(x))\right] - \alpha \frac{\partial \tilde{y}^*(x)}{\partial x} \tilde{y}^*(x)$$

$$= \nabla_x g(x, y) - \nabla_x g(x, \tilde{y}^*(x)) - \frac{\partial \tilde{y}^*(x)}{\partial x} \left[\nabla_y g(x, \tilde{y}^*(x)) + \alpha \tilde{y}^*(x)\right].$$

The first order optimality condition ensures that $\nabla_y g(x, \tilde{y}^*(x)) + \alpha \tilde{y}^*(x) = 0$. Hence, we obtain the desired result:

$$\nabla_x \tilde{h}(x, y) = \nabla_x g(x, y) - \nabla_x g(x, \tilde{y}^*(x)).$$

$\square$

## F  Relationship between the KKT Points of Equations (2) and (3)

In this section, we prove the connection between the KKT points of eqs. (2) and (3).

To this end, we first note that $g^*(z)$ is only continuous and in general not differentiable. Thus, we first give the formal definition of the subdifferential, which is commonly used in the definition of KKT points for non-differentiable functions Ma et al. (2020). The subdifferential is defined as follows.

**Definition 2.** *Given a continuous function $\omega(x)$ defined on $\mathcal{X} \subseteq \mathbb{R}^n$ and $x \in \mathcal{X}$, a vector $v \in \mathbb{R}^n$ is a subdifferential of $\omega(x)$ at $x$ if and only if*

$$\lim_{x' \to x} \omega(x') - \omega(x) \geq \langle v, x' - x \rangle + o(\|x' - x\|_2).$$

*We denote $\partial \omega(x)$ as the set of all subdifferentials at $x$.*

We next characterize a subdifferential of the function $h(z)$.

**Lemma 2.** *A subdifferential of $h(z)$ is given as follows. For all $z \in \mathcal{Z}$, we have*

$$(\nabla_x g(z) - \nabla_x g(x, y^*(x))), \nabla_y g(z))^\top \in \partial h(z).$$

*Proof of Lemma 2.* Since $h(z) := g(z) - g^*(z) = g(z) - g^*(x)$ and $g(z)$ is differentiable, a subdifferential of $h(z)$ will readily follow if we obtain a subdifferential of $-g^*(x)$.

We next show that if Assumption 1 holds, given any fixed $\bar{x} \in \mathcal{X}$, a subdifferential of $-g^*(\bar{x})$ is given by

$$-\nabla_x g(\bar{x}, y^*(\bar{x})) \in \partial(-g^*(\bar{x})),$$

where $y^*(\bar{x})$ is any point satisfying $y^*(\bar{x}) \in \arg\min_{y \in \mathbb{R}^d} g(\bar{x}, y)$.

To prove the above statement, fix a $\bar{x} \in \mathcal{X}$, and let $J(x) := -g^*(x) + \frac{\rho \|x - \bar{x}\|_2^2}{2}$, where $\bar{x}$ is an arbitrary point inside $\mathcal{X}$, $g^*(x) = \min_{y \in \mathbb{R}^d} g(x, y)$, and $\rho \geq \rho_g$ is a constant. We next prove that $J(x)$ is a $\rho - \rho_g$ strongly convex function.

For any $x, \tilde{x} \in \mathcal{X}$, and $0 \leq \lambda \leq 1$, we have

$$J(\lambda x + (1 - \lambda)\tilde{x}) - \lambda J(x) - (1 - \lambda)J(\tilde{x})$$
$$= -g^*(\lambda x + (1 - \lambda)\tilde{x}) + \frac{\rho \|\lambda x + (1 - \lambda)\tilde{x} - \bar{x}\|_2^2}{2} + \lambda g^*(x) - \frac{\lambda \rho \|x - \bar{x}\|_2^2}{2} + \lambda g^*(\tilde{x}) - \frac{\lambda \rho \|\tilde{x} - \bar{x}\|_2^2}{2}$$
$$= \lambda[g^*(x) - g^*(\lambda x + (1 - \lambda)\tilde{x})] + (1 - \lambda)[g^*(\tilde{x}) - g^*(\lambda x + (1 - \lambda)\tilde{x})]$$
$$\quad + \frac{\rho}{2}\left[\|\lambda x + (1 - \lambda)\tilde{x}\|_2^2 - \lambda\|x\|_2^2 - (1 - \lambda)\|\tilde{x}\|_2^2\right]$$
$$= \lambda[g^*(x) - g^*(\lambda x + (1 - \lambda)\tilde{x})] + (1 - \lambda)[g^*(\tilde{x}) - g^*(\lambda x + (1 - \lambda)\tilde{x})] - \frac{\rho \lambda(1 - \lambda)\|x - \tilde{x}\|_2^2}{2}. \quad (16)$$

Recall the definition $\mathcal{S}_x := \arg\min_{y \in \mathbb{R}^d} g(x, y)$. Let $u \in \mathcal{S}_{\lambda x + (1-\lambda)\tilde{x}}$. Then $g^*(\lambda x + (1 - \lambda)\tilde{x}) = g(\lambda x + (1 - \lambda)\tilde{x}, u)$, and by the definition of $g^*$, we have $g^*(x) \leq g(x, u)$ and $g^*(\tilde{x}) \leq g(\tilde{x}, u)$. These three relationships together with eq. (16) yield

$$J(\lambda x + (1 - \lambda)\tilde{x}) - \lambda J(x) - (1 - \lambda)J(\tilde{x})$$
$$\leq \lambda[g(x, u) - g(\lambda x + (1 - \lambda)\tilde{x}, u)] + (1 - \lambda)[g(\tilde{x}, u) - g(\lambda x + (1 - \lambda)\tilde{x}, u)]$$
$$\quad - \frac{\rho \lambda(1 - \lambda)\|x - \tilde{x}\|_2^2}{2}. \quad (17)$$

Recall that the $\rho_g$ Lipschitz gradient of $g$ indicates the following two inequalities.

$$g(x, u) - g(\lambda x + (1 - \lambda)\tilde{x}, u)$$
$$\leq \langle \nabla_x g(\lambda x + (1 - \lambda)\tilde{x}, u), x - (\lambda x + (1 - \lambda)\tilde{x}) \rangle + \frac{\rho_g \|x - \lambda x - (1 - \lambda)\tilde{x}\|_2^2}{2}$$
$$= (1 - \lambda) \langle \nabla_x g(\lambda x + (1 - \lambda)\tilde{x}, u), x - \tilde{x} \rangle + \frac{\rho_g (1 - \lambda)^2 \|x - \tilde{x}\|_2^2}{2}. \tag{18}$$

$$g(\tilde{x}, u) - g(\lambda x + (1 - \lambda)\tilde{x}, u)$$
$$\leq \langle \nabla_x g(\lambda x + (1 - \lambda)\tilde{x}, u), \tilde{x} - (\lambda x + (1 - \lambda)\tilde{x}) \rangle + \frac{\rho_g \|\tilde{x} - \lambda x - (1 - \lambda)\tilde{x}\|_2^2}{2}$$
$$= -\lambda \langle \nabla_x g(\lambda x + (1 - \lambda)\tilde{x}, u), x - \tilde{x} \rangle + \frac{\rho_g \lambda^2 \|x - \tilde{x}\|_2^2}{2}. \tag{19}$$

Substituting eqs. (18) and (19) into eq. (17) yields

$$J(\lambda x + (1 - \lambda)\tilde{x}) \leq \lambda J(x) + (1 - \lambda)J(\tilde{x}) - \frac{(\rho - \rho_g)(1 - \lambda)\lambda \|x - \tilde{x}\|_2^2}{2},$$

which is exactly the sufficient and necessary condition of $(\rho - \rho_g)$-strongly convex.

Next, we prove that $\partial(-g^*)(\bar{x}) = \partial J(\bar{x})$. Suppose $v \in \partial J(\bar{x})$. Then, we have

$$\langle v, x - \bar{x} \rangle + o(\|x - \bar{x}\|_2) \leq J(x) - J(\bar{x}) = -g^*(x) + \frac{\rho \|x - \bar{x}\|_2^2}{2} + g^*(\bar{x})$$
$$= -g^*(x) + g^*(\bar{x}) + o(\|x - \bar{x}\|_2).$$

The above inequality implies that $-g^*(x) + g^*(\bar{x}) \geq \langle v, x - \bar{x} \rangle + o(\|x - \bar{x}\|_2)$, i.e., $v \in \partial(-g^*)(\bar{x})$.

Furthermore, suppose $v \in \partial(-g^*)(\bar{x})$ and $J(x) = -g^*(x) + \frac{\rho \|x - \bar{x}\|_2^2}{2}$. By Definition 2, we have

$$\langle v, x - \bar{x} \rangle + o(\|x - \bar{x}\|_2) \leq -g^*(x) + g^*(\bar{x}) = J(x) - J(\bar{x}) - \frac{\rho \|x - \bar{x}\|_2^2}{2}.$$

The above inequality implies that $J(x) - J(\bar{x}) \geq \langle v, x - \bar{x} \rangle + o(\|x - \bar{x}\|_2)$, i.e., $v \in \partial J(\bar{x})$.

Moreover, we show that $-\nabla g(\bar{x}, y^*(\bar{x}))$ is a subdifferential of $J(x)$. Following from the definition of $g^*(x)$, we have $g^*(x) \leq g(x, y^*(\bar{x}))$, and thus the following inequality holds

$$g^*(x) - g^*(\bar{x}) \leq g(x, y^*(\bar{x})) - g^*(\bar{x}) = g(x, y^*(\bar{x})) - g(\bar{x}, y^*(\bar{x})). \tag{20}$$

Recall the gradient Lipschitz condition of $g$ with $y$ fixed at $y^*(\bar{x})$, and then we have

$$g(x, y^*(\bar{x})) \leq g(\bar{x}, y^*(\bar{x})) + \langle \nabla_x g(\bar{x}, y^*(\bar{x})), x - \bar{x} \rangle + \frac{\rho_g \|x - \bar{x}\|_2}{2}$$
$$\leq g(\bar{x}, y^*(\bar{x})) + \langle \nabla_x g(\bar{x}, y^*(\bar{x})), x - \bar{x} \rangle + \frac{\rho \|x - \bar{x}\|_2}{2}. \tag{21}$$

Substituting eq. (21) into eq. (20) yields

$$J(x) = -g^*(x) + \frac{\rho \|x - \bar{x}\|_2^2}{2} \geq -g^*(\bar{x}) - \langle \nabla_x g(\bar{x}, y^*(\bar{x})), x - \bar{x} \rangle = J(\bar{x}) + \langle -\nabla_x g(\bar{x}, y^*(\bar{x})), x - \bar{x} \rangle.$$

Since the above inequality holds for all $x \in \mathcal{X}$, $-\nabla_x g(\bar{x}, y^*(\bar{x}))$ is the subdifferential of $J(x)$ at $\bar{x}$, and thus is the subdifferential of $-g^*(x)$.

With $-\nabla_x g(\bar{x}, y^*(\bar{x}))$ being characterized as the subdifferential of $-g^*(x)$, the subdifferential of $h(z) := g(z) - g^*(z)$ w.r.t. $x$ can then be obtained immediately by adding $\nabla_x g(z)$ and $\partial(-g^*(x))$. Together with the fact that $\nabla_y h(z) = \nabla_y g(z)$, the subdifferential $\partial h(z)$ with respect to $z$ is given by $\partial h(z) = (\nabla_x g(x, y) - \nabla_x g(x, y^*(x))), \nabla_y g(x, y))^\top$. □

Then the KKT points of eq. (2) can be defined in the same way as Definition 1 except that we replace $h(z)$ and $\partial h(z)$ with $\tilde{h}(z)$ and $\nabla \tilde{h}(z)$ respectively. With the well-defined KKT points of eq. (2), we next characterize the connections of the KKT points of Equations (2) and (3) below.

**Proposition 2** (Re-statement of Proposition 1). *Suppose Assumption 1 holds. If $z_{\hat{k}} = (x_{\hat{k}}, y_{\hat{k}})$ is an $\epsilon$-KKT point of the problem in eq. (3), it is also an $\tilde{\epsilon}$-KKT point of eq. (2) with $\tilde{\epsilon} = \mathcal{O}(\epsilon + \alpha + \delta)$.*

Clearly, if we set $\alpha = \mathcal{O}(\epsilon)$ and $\delta = \mathcal{O}(\epsilon)$, then we also obtain an $\epsilon$-KKT point of eq. (2).

*Proof.* The definition of $\epsilon$-KKT point ensures that there exists $\bar{z} = (\bar{x}, \bar{y})$ with $\|\bar{z} - z_{\hat{k}}\|^2 \leq \epsilon$ and $\bar{\lambda} \geq 0$ such that

$$\left| \bar{\lambda} \tilde{h}(\bar{z}) \right| \leq \epsilon \tag{22}$$

$$\text{dist} \left( \nabla f(\bar{z}) + \bar{\lambda} \nabla \tilde{h}(\bar{z}) + \mathcal{N}(z; \mathcal{Z}), 0 \right)^2 \leq \epsilon. \tag{23}$$

Replacing the expression of $\tilde{h}(\bar{z})$ by its definition in eq. (22), we obtain

$$\left| \bar{\lambda} g(\bar{x}, \bar{y}) - \bar{\lambda} \tilde{g}^*(\bar{x}) - \bar{\lambda} \delta \right| \leq \epsilon \tag{24}$$

which further implies

$$\left| \bar{\lambda} g(\bar{x}, \bar{y}) - \bar{\lambda} g^*(\bar{x}) \right| - \left| \bar{\lambda} \tilde{g}^*(\bar{x}) - \bar{\lambda} g^*(\bar{x}) \right| - \bar{\lambda} \delta \leq \epsilon \tag{25}$$

due to the triangle inequality. Therefore, we obtain

$$\begin{aligned} \bar{\lambda} \Big[ g(\bar{x}, \bar{y}) - g^*(\bar{x}) \Big] &\leq \bar{\lambda} \left| \tilde{g}^*(\bar{x}) - g^*(\bar{x}) \right| + \bar{\lambda} \delta + \epsilon \\ &\leq D \left| \tilde{g}^*(\bar{x}) - g^*(\bar{x}) \right| + D\delta + \epsilon \end{aligned} \tag{26}$$

Further, we have

$$\begin{aligned} \left| \tilde{g}^*(\bar{x}) - g^*(\bar{x}) \right| &= \left| \min_y \tilde{g}(\bar{x}, y) - \min_y g(\bar{x}, y) \right| \\ &\leq \max_y \left| \tilde{g}(\bar{x}, y) - g(\bar{x}, y) \right| = \max_y \frac{\alpha}{2} \|y\|^2 = \frac{\alpha}{2} D_z^2. \end{aligned} \tag{27}$$

Combining eq. (26) and eq. (27) yields

$$\bar{\lambda} \Big[ g(\bar{x}, \bar{y}) - g^*(\bar{x}) \Big] \leq \frac{\alpha D}{2} D_z^2 + D\delta + \epsilon. \tag{28}$$

Suppose $v \in \mathcal{N}(z; \mathcal{Z})$ is the vector that attains the minimum of the following problem

$$\min_{u \in \mathcal{N}(\bar{z}; \mathcal{Z})} \|u + \nabla f(\bar{z}) + \bar{\lambda} \nabla \tilde{h}(\bar{z})\|_2.$$

This implies that

$$\begin{aligned} &\text{dist} \left( \nabla f(\bar{z}) + \bar{\lambda} \nabla \tilde{h}(\bar{z}) + \mathcal{N}(z; \mathcal{Z}), 0 \right)^2 \\ &= \left\| v_x + \nabla_x f(\bar{x}, \bar{y}) + \bar{\lambda} \nabla_x \tilde{h}(\bar{x}, \bar{y}) \right\|^2 + \left\| v_y + \nabla_y f(\bar{x}, \bar{y}) + \bar{\lambda} \nabla_y \tilde{h}(\bar{x}, \bar{y}) \right\|^2 \end{aligned}$$

where $v = (v_x, v_y)$. Hence eq. (23) implies

$$\underbrace{\left\| v_x + \nabla_x f(\bar{x}, \bar{y}) + \bar{\lambda} \Big[ \nabla_x g(\bar{x}, \bar{y}) - \nabla_x g(\bar{x}, \tilde{y}^*(\bar{x})) \Big] \right\|}_{\tilde{P}_x}^2 + \underbrace{\left\| v_y + \nabla_y f(\bar{x}, \bar{y}) + \bar{\lambda} \nabla_y g(\bar{x}, \bar{y}) \right\|}_{P_y}^2 \leq \epsilon \tag{29}$$

Let $P_x = v_x + \nabla_x f(\bar{x}, \bar{y}) + \bar{\lambda}\Big[\nabla_x g(\bar{x}, \bar{y}) - \nabla_x g(\bar{x}, y^*(\bar{x}))\Big]$. We have

$$\left\|P_x\right\|^2 + \left\|P_y\right\|^2 = \left\|P_x\right\|^2 - \left\|\tilde{P}_x\right\|^2 + \left\|\tilde{P}_x\right\|^2 + \left\|P_y\right\|^2 \overset{(29)}{\leq} \left\|P_x\right\|^2 - \left\|\tilde{P}_x\right\|^2 + 2\epsilon. \tag{30}$$

Also we can derive

$$\left|\left\|P_x\right\|^2 - \left\|\tilde{P}_x\right\|^2\right| = \left|\langle P_x + \tilde{P}_x, P_x - \tilde{P}_x\rangle\right| \leq \left\|P_x + \tilde{P}_x\right\|\left\|P_x - \tilde{P}_x\right\|$$

$$= \bar{\lambda}\left\|2v_x + 2\nabla_x f(\bar{x}, \bar{y}) + 2\bar{\lambda}\nabla_x g(\bar{x}, \bar{y}) - \bar{\lambda}\nabla_x g(\bar{x}, y^*(\bar{x})) - \bar{\lambda}\nabla_x g(\bar{x}, \tilde{y}^*(\bar{x}))\right\|$$

$$\times \left\|\nabla_x g(\bar{x}, y^*(\bar{x})) - \nabla_x g(\bar{x}, \tilde{y}^*(\bar{x}))\right\|$$

$$\overset{(i)}{\leq} 6DL_g(L_f + 2DL_g)\|\tilde{y}^*(\bar{x}) - y^*(\bar{x})\|,$$

where $(i)$ follows because $\|v_x\|_2 \leq 2\|\nabla_x f(\bar{x}, \bar{y}) + \bar{\lambda}\nabla_x \tilde{h}(\bar{x}, \bar{y})\|_2$, which can be easily proved by contradictory. Hence, using the fact that $\|\tilde{y}^*(\bar{x}) - y^*(\bar{x})\| \leq \mathcal{O}(\alpha)$, we obtain

$$\left|\left\|P_x\right\|^2 - \left\|\tilde{P}_x\right\|^2\right| \leq \epsilon + \mathcal{O}(\alpha),$$

which in conjunction with eq. (30) implies

$$\left\|v_x + \nabla_x f(\bar{x}, \bar{y}) + \bar{\lambda}\Big[\nabla_x g(\bar{x}, \bar{y}) - \nabla_x g(\bar{x}, y^*(\bar{x}))\Big]\right\|^2 + \left\|v_y + \nabla_y f(\bar{x}, \bar{y}) + \bar{\lambda}\nabla_y g(\bar{x}, \bar{y})\right\|^2$$

$$\leq \mathcal{O}(\epsilon + \alpha).$$

This is equivalent to

$$\left\|v + \begin{pmatrix} \nabla_x f(\bar{x}, \bar{y}) + \bar{\lambda}\Big[\nabla_x g(\bar{x}, \bar{y}) - \nabla_x g(\bar{x}, y^*(\bar{x}))\Big] \\ \nabla_y f(\bar{x}, \bar{y}) + \bar{\lambda}\nabla_y g(\bar{x}, \bar{y}) \end{pmatrix}\right\|_2 \leq \mathcal{O}(\epsilon + \alpha).$$

The above inequality implies

$$\text{dist}\left(\begin{pmatrix} \nabla_x f(\bar{x}, \bar{y}) + \bar{\lambda}\Big[\nabla_x g(\bar{x}, \bar{y}) - \nabla_x g(\bar{x}, y^*(\bar{x}))\Big] \\ \nabla_y f(\bar{x}, \bar{y}) + \bar{\lambda}\nabla_y g(\bar{x}, \bar{y}) \end{pmatrix} + \mathcal{N}(z; \mathcal{Z}), 0\right)^2 \leq \mathcal{O}(\epsilon + \alpha). \tag{31}$$

Since $v \in \mathcal{N}(\tilde{z}, \mathcal{Z})$, eqs. (28) and (31) together fulfill the KKT condition, and thus complete the proof. $\quad\square$

## G   Proof of Theorem 1

### G.1   Supporting Lemmas

We first cite two standard lemmas, which are useful for our proof here.

**Lemma 3** (Lemma 3.5 Lan (2020)). *Suppose that $\mathcal{S}$ is a convex and closed subset of $\mathbb{R}^n$, $x \in \mathcal{S}$, and $v \in \mathbb{R}^n$. Define $\bar{x} = \Pi_{\mathcal{S}}(x - v)$. Then, for any $\tilde{x} \in \mathcal{S}$, the following inequality holds:*

$$\langle x, v\rangle + \tfrac{1}{2}\|\bar{x} - \tilde{x}\|_2^2 + \tfrac{1}{2}\|x - \bar{x}\|_2^2 \leq \tfrac{1}{2}\|x - \tilde{x}\|_2^2.$$

**Lemma 4** (Theorem 2.2.14 Nesterov et al. (2018)). *Suppose that Assumption 1 holds. Consider the projected gradient descent in eq. (5). Define $\tilde{y}^*(x_t) \coloneqq \arg\min_{y \in \mathbb{R}^d} g(x_t, y) + \frac{\alpha}{2}\|y\|_2^2$. We have*

$$\|\hat{y}^*(x_t) - \tilde{y}^*(x_t)\|_2 = \|\hat{y}_N - \tilde{y}^*(x_t)\|_2 \leq \left(1 - \frac{\alpha}{\rho_g + 2\alpha}\right)^N \|\hat{y}_0 - \tilde{y}^*(x_t)\|_2.$$

Next, we establish an upper bound on the optimal dual variable in the following lemma.

**Lemma 5.** *Suppose that Assumptions 1 and 2 hold. Then, there exists $\lambda^*$ satisfying $0 \leq \lambda^* \leq \frac{D_f}{\delta}$, where $D_f = \sup_{z,z' \in \mathcal{Z}} |f(z) - f(z')|$, so that for $z^* := argmin_{z \in \mathcal{Z}} f(z) + \lambda^* \tilde{h}(z)$, $\tilde{h}(z^*) \leq 0$, and the KKT condition holds, i.e., $\lambda^* \tilde{h}(z^*) = 0$, and $\nabla f(z^*) + \lambda^* \nabla \tilde{h}(z^*) \in -\mathcal{N}_{\mathcal{Z}}(z^*)$, where $\mathcal{N}_{\mathcal{Z}}(z^*)$ is the normal cone defined as $\mathcal{N}_{\mathcal{Z}}(z^*) = \{v \in \mathbb{R}^{p+d} : \langle v, z - z^* \rangle \leq 0, \text{ for all } z \in \mathcal{Z}\}$.*

*Proof.* Pick any $x_0 \in \mathcal{X}$. Let $y_0 = \arg\min_{y \in \mathbb{R}^d} g(x,y) + \frac{\alpha}{2}\|y\|_2^2$, and $z_0 = (x_0, y_0)$. Then, $\tilde{h}(z_0) = g(x_0, y_0) - \tilde{g}^*(x_0) - \delta = -\delta$ holds, which implies that $z_0$ is a strictly feasible point. The existence of such a strictly feasible point $z_0$ ensures that the Slater's condition holds, and then the standard result (see, e.g., Lan (2020)) implies the existence of $\lambda^* \geq 0$ that satisfies for $z^* := argmin_{z \in \mathcal{Z}} f(z) + \lambda^* \tilde{h}(z)$, $\tilde{h}(z^*) \leq 0$, $\lambda^* \tilde{h}(z^*) = 0$, and $\nabla f(z^*) + \lambda^* \nabla \tilde{h}(z^*) \in -\mathcal{N}_{\mathcal{Z}}(z^*)$.

Define the dual function as $d(\lambda) := \min_{z \in \mathcal{Z}} \mathcal{L}(z, \lambda)$. Then, we have, for any $\lambda$ and $z \in \mathcal{Z}$,

$$d(\lambda) \leq f(z_0) + \lambda \hat{h}(z_0) = f(z_0) - \delta\lambda. \tag{32}$$

Taking $\lambda = \lambda^*$ in eq. (32) and using the fact that $|d(\lambda^*) - f(z_0)| = |f(z^*) - f(z_0)| \leq D_f$, where $D_f = \sup_{z,z' \in \mathcal{Z}} |f(z) - f(z')|$, we complete the proof. □

In the next lemma, we show that the constrained function $\tilde{h}(z)$ is gradient Lipschitz continuous.

**Lemma 6.** *Suppose that Assumption 1 holds. Then, the gradient $\nabla \tilde{h}(z) = \left(\nabla_x \tilde{h}(x,y), \nabla_y \tilde{h}(x,y)\right)^\top$ is Lipschitz continuous with constant $\rho_h = \rho_g(2 + \rho_g/\alpha)$.*

*Proof.* Recall the form of $\nabla \tilde{h}(z)$ is given by

$$\nabla \tilde{h}(z) = \begin{pmatrix} \nabla_x g(x,y) \\ \nabla_y g(x,y) \end{pmatrix} - \begin{pmatrix} \nabla_x g(x, \tilde{y}^*(x)) \\ \mathbf{0}_d \end{pmatrix} = \nabla g(z) - \begin{pmatrix} G_x \\ G_y \end{pmatrix},$$

where $G_x := \nabla_x g(x, \tilde{y}^*(x))$ and $G_y := \mathbf{0}_d \in \mathbb{R}^d$ is a vector of all zeros. Taking derivative w.r.t. $z$ yields:

$$\begin{aligned} \nabla^2 \tilde{h}(z) =& \nabla^2 g(z) - \begin{pmatrix} \frac{\partial G_x}{\partial x} & \frac{\partial G_x}{\partial y} \\ \frac{\partial G_y}{\partial x} & \frac{\partial G_y}{\partial y} \end{pmatrix} \\ =& \nabla^2 g(z) - \underbrace{\begin{pmatrix} \frac{\partial \nabla_x g(x, \tilde{y}^*(x))}{\partial x} & \mathbf{0}_{p \times d} \\ \mathbf{0}_{d \times p} & \mathbf{0}_{d \times d} \end{pmatrix}}_{M}. \end{aligned} \tag{33}$$

where $\mathbf{0}_{m \times n} \in \mathbb{R}^{m \times n}$ is a matrix of all zeros.

Recall that in Assumption 1, we assume $\|\nabla g(z) - \nabla g(z')\|_2 \leq \rho_g\|z - z'\|_2$ for all $z, z' \in \mathcal{Z}$, which immediately implies $\|\nabla^2 g(z)\|_2 \leq \rho_g$. Note that in the sequel, $\|\cdot\|_2$ of a matrix denotes the spectral norm. Moreover, let

$$\nabla^2 g(z) = \begin{pmatrix} \nabla^2_{xx} g(z) & \nabla^2_{xy} g(z) \\ \nabla^2_{yx} g(z) & \nabla^2_{yy} g(z) \end{pmatrix}.$$

Given any $x \in \mathbb{R}^p$, we have $\nabla^2 g(z) \begin{pmatrix} x \\ 0 \end{pmatrix} = \begin{pmatrix} \nabla^2_{xx} g(z) x \\ \nabla^2_{yx} g(z) x \end{pmatrix}$. Thus, the following inequality holds

$$\|\nabla^2_{xx} g(z) x\|_2 \leq \left\| \begin{pmatrix} \nabla^2_{xx} g(z) x \\ \nabla^2_{yx} g(z) x \end{pmatrix} \right\|_2 = \left\| \nabla^2 g(z) \begin{pmatrix} x \\ 0 \end{pmatrix} \right\|_2 \leq \|\nabla^2 g(z)\|_2 \|x\|_2 \leq \rho_g \|x\|_2. \tag{34}$$

Following the definition of the spectral norm of $\nabla_{xx}^2 g(z)$, we have

$$\|\nabla_{xx}^2 g(z)\|_2 \le \rho_g. \tag{35}$$

Following the similar steps to eq. (34), we have

$$\|\nabla_{yx}^2 g(z)\|_2 \le \rho_g. \tag{36}$$

We next upper-bound the spectral norm of the matrix $M$ defined in eq. (33). We have:

$$\|\nabla^2 \tilde{g}^*(z)\|_2 = \|M\|_2 = \left\| \begin{pmatrix} \frac{\partial \nabla_x g(x, \tilde{y}^*(x))}{\partial x} & \mathbf{0}_{p \times d} \\ \mathbf{0}_{d \times p} & \mathbf{0}_{d \times d} \end{pmatrix} \right\|_2 \le \left\| \frac{\partial \nabla_x g(x, \tilde{y}^*(x))}{\partial x} \right\|_2, \tag{37}$$

where eq. (37) follows from the fact that for the block matrix $C = \begin{pmatrix} A & \mathbf{0} \\ \mathbf{0} & B \end{pmatrix}$, we have $\|C\|_2 \le \max\{\|A\|_2, \|B\|_2\}$. Further, using the chain rule, we obtain

$$\frac{\partial \nabla_x g(x, \tilde{y}^*(x))}{\partial x} = \nabla_x^2 g(x, \tilde{y}^*(x)) + \frac{\partial \tilde{y}^*(x)}{\partial x} \nabla_y \nabla_x g(x, \tilde{y}^*(x)). \tag{38}$$

Thus, taking the norm on both sides of the above equation and applying the triangle inequality yield

$$\begin{aligned} \left\| \frac{\partial \nabla_x g(x, \tilde{y}^*(x))}{\partial x} \right\|_2 &\le \left\| \nabla_{xx}^2 g(x, \tilde{y}^*(x)) \right\|_2 + \left\| \frac{\partial \tilde{y}^*(x)}{\partial x} \right\|_2 \left\| \nabla_y \nabla_x g(x, \tilde{y}^*(x)) \right\|_2 \\ &\overset{(i)}{\le} \rho_g + \rho_g \left\| \frac{\partial \tilde{y}^*(x)}{\partial x} \right\|_2, \end{aligned} \tag{39}$$

where $(i)$ follows from eqs. (35) and (36).

Applying implicit differentiation w.r.t. $x$ to the optimality condition of $\tilde{y}^*(x)$ implies $\nabla_y g(x, \tilde{y}^*(x)) + \alpha \tilde{y}^*(x) = 0$. This yields

$$\nabla_x \nabla_y g(x, \tilde{y}^*(x)) + \frac{\partial \tilde{y}^*(x)}{\partial x} \nabla_y^2 g(x, \tilde{y}^*(x)) + \alpha \frac{\partial \tilde{y}^*(x)}{\partial x} = 0,$$

which further yields

$$\frac{\partial \tilde{y}^*(x)}{\partial x} = -\left[ \nabla_y^2 g(x, \tilde{y}^*(x)) + \alpha \mathbf{I} \right]^{-1} \nabla_x \nabla_y g(x, \tilde{y}^*(x)).$$

Hence, we obtain

$$\left\| \frac{\partial \tilde{y}^*(x)}{\partial x} \right\|_2 \le \left\| \left[ \nabla_y^2 g(x, \tilde{y}^*(x)) + \alpha \mathbf{I} \right]^{-1} \right\|_2 \left\| \nabla_x \nabla_y g(x, \tilde{y}^*(x)) \right\|_2 \le \frac{\rho_g}{\alpha}, \tag{40}$$

where the last inequality follows from Assumption 1. Hence, combining eq. (37), eq. (39), and eq. (40), we have

$$\|\nabla^2 \tilde{g}^*(z)\|_2 = \|M\|_2 \le \rho_g + \rho_g \frac{\rho_g}{\alpha}, \tag{41}$$

which, in conjunction of eq. (33), yields

$$\left\| \nabla^2 \tilde{h}(z) \right\|_2 \le \left\| \nabla^2 g(z) \right\|_2 + \left\| M \right\|_2 \le \rho_g + \rho_g + \frac{\rho_g^2}{\alpha} = \rho_g \left( 2 + \frac{\rho_g}{\alpha} \right).$$

This completes the proof. $\qquad\square$

### G.2 Proof of Theorem 1

Based on the above lemmas, we develop the proof of Theorem 1. We first formally restate the theorem with the full details.

**Theorem 3** (Formal Statement of Theorem 1). *Suppose Assumption 1 holds. Let $\gamma_t = t + t_0 + 1$, $\eta_t = \frac{\mu(t+t_0+1)}{2}$, $\tau_t = \frac{4L_g^2}{\mu t}$, $\theta_t = \frac{t+t_0}{t+t_0+1}$, where $L_g = \sup_z \|\nabla_z g(z)\|_2$, $t_0 = \frac{2(\rho_f + B\rho_h)}{\mu}$, $B = \frac{D_f}{\delta} + 1$, where $D_f = \sup_{z,z' \in \mathcal{Z}} |f(z) - f(z')|$, and $\rho_h$ is given in Lemma 6. Then, we have*

$$f(\bar{z}) - f(z^*) \leq \frac{2L_g BD_{\mathcal{Z}}}{T^2} + \frac{\gamma_0(\eta_0 - \mu)\|z^* - z_0\|_2^2}{T^2} + (\rho_g D_{\mathcal{Z}} + 4L_g)BD_{\mathcal{Z}}\left(1 - \frac{\alpha}{\rho_g + 2\alpha}\right)^N,$$

$$[\tilde{h}(\bar{z})]_+ \leq \frac{2L_g BD_{\mathcal{Z}} + \gamma_0 \tau_0 B^2 + \gamma_0(\eta_0 - \mu)D_{\mathcal{Z}}^2}{T^2} + (\rho_g D_{\mathcal{Z}} + 4L_g)BD_{\mathcal{Z}}\left(1 - \frac{\alpha}{\rho_g + 2\alpha}\right)^N,$$

*and*

$$\|\bar{z} - z^*\|_2^2 \leq \frac{2\gamma_0 \tau_0 B^2 + 2\gamma_0(\eta_0 - \mu)D_{\mathcal{Z}}^2}{\mu T^2} + \frac{2(T + t_0 + 1)^2}{T^2}(\rho_g D_{\mathcal{Z}} + 4L_g)BD_{\mathcal{Z}}\left(1 - \frac{\alpha}{\rho_g + 2\alpha}\right)^N,$$

*where $z^* = \arg\min_{z \in \mathcal{Z}}\{f(z) : \tilde{h}(z) \leq 0\}$ and $D_{\mathcal{Z}} = \max\{\sqrt{\|x - x'\|_2^2 + \|y - y'\|_2^2} : x, x' \in \mathcal{X}, y \in \mathcal{S}_x, y' \in \mathcal{S}_{x'}\}$.*

*Proof.* We first define some notations that will be used later. Let $\hat{d}_t = (1 + \theta_t)\hat{h}(z_t) - \theta_t \hat{h}(z_{t-1})$, $d_t = (1 + \theta_t)\tilde{h}(z_t) - \theta_t \tilde{h}(z_{t-1})$, and $\xi_t = \hat{h}(z_t) - \hat{h}(z_{t-1})$. Furthermore, we define the primal-dual gap function as

$$Q(w, \tilde{w}) \coloneqq f(z) + \tilde{\lambda}\tilde{h}(z) - \left(f(\tilde{z}) + \lambda\tilde{h}(\tilde{z})\right),$$

where $w = (z, \lambda)$, $\tilde{w} = (\tilde{z}, \tilde{\lambda}) \in \mathcal{Z} \times \Lambda$ are primal-dual pairs.

Consider the update of $\lambda$ in eq. (6). Applying Lemma 3 with $v = -\hat{d}_t/\tau_t$, $\mathcal{S} = \Lambda$, $\bar{x} = \lambda_{t+1}$, $x = \lambda_t$ and letting $\tilde{x} = \lambda$ be an arbitrary point inside $\Lambda$, we have

$$-(\lambda_{t+1} - \lambda)\hat{d}_t \leq \frac{\tau_t}{2}\left((\lambda - \lambda_t)^2 - (\lambda_{t+1} - \lambda_t)^2 - (\lambda - \lambda_{t+1})^2\right). \tag{42}$$

Similarly, consider the update of $z$ in eq. (7). Applying Lemma 3 with

$$v = \frac{1}{\eta_t}\left(\nabla f(z_t) + \lambda_{t+1}\hat{\nabla}\tilde{h}(z_t)\right) \coloneqq \frac{1}{\eta_t}\hat{\nabla}\mathcal{L}(z_t, \lambda_{t+1}),$$

$\mathcal{S} = \mathcal{Z}$, $\bar{x} = z_{t+1}$, $x = z_t$ and let $\tilde{x} = z$ be an arbitrary point inside $\mathcal{Z}$, we obtain

$$\langle \hat{\nabla}_z \mathcal{L}(z_t, \lambda_{t+1}), z_{t+1} - z \rangle \leq \frac{\eta_t}{2}\left((z - z_t)^2 - (z_{t+1} - z_t)^2 - (z - z_{t+1})^2\right). \tag{43}$$

Recall that $f(z)$ and $\tilde{h}(z)$ are $\rho_f$- and $\rho_h$-gradient Lipschitz (see Assumption 1 and Lemma 6). This implies

$$\langle \nabla f(z_t), z_{t+1} - z_t \rangle \geq f(z_{t+1}) - f(z_t) - \frac{\rho_f \|z_t - z_{t+1}\|_2^2}{2}, \tag{44}$$

$$\langle \nabla \tilde{h}(z_t), z_{t+1} - z_t \rangle \geq \tilde{h}(z_{t+1}) - \tilde{h}(z_t) - \frac{\rho_h \|z_t - z_{t+1}\|_2^2}{2}. \tag{45}$$

Moreover, Assumption 2 assumes $f(z)$ is a $\mu$-strongly convex function, which yields

$$\langle \nabla f(z_t), z_t - z \rangle \geq f(z_t) - f(z) + \frac{\mu\|z - z_t\|_2^2}{2}. \tag{46}$$

The convexity of $\tilde{h}(z)$ in Assumption 2 ensures that

$$\langle \nabla \tilde{h}(z_t), z_t - z \rangle \geq \tilde{h}(z_t) - \tilde{h}(z). \tag{47}$$

For the exact gradient of Lagrangian with respect to the primal variable, we have

$$\begin{aligned}
&\langle \nabla_z \mathcal{L}(z_t, \lambda_{t+1}), z_{t+1} - z \rangle \\
&= \langle \nabla f(z_t) + \lambda_{t+1} \nabla \tilde{h}(z_t), z_{t+1} - z \rangle \\
&= \langle \nabla f(z_t), z_{t+1} - z_t \rangle + \langle \nabla f(z_t), z_t - z \rangle + \lambda_{t+1} \langle \nabla \tilde{h}(z_t), z_{t+1} - z_t \rangle + \lambda_{t+1} \langle \nabla \tilde{h}(z_t), z_t - z \rangle \\
&\overset{(i)}{\geq} f(z_{t+1}) - f(z) + \lambda_{t+1}(\tilde{h}(z_{t+1}) - \tilde{h}(z)) - \frac{\rho_f + \lambda_{t+1}\rho_h \|z_{t+1} - z_t\|_2^2}{2} + \frac{\mu\|z - z_t\|_2^2}{2},
\end{aligned} \tag{48}$$

where $(i)$ follows from eqs. (44) to (47).

Combining eqs. (43) and (48) yields

$$\begin{aligned}
f(z_{t+1}) - f(z) \leq{}& \langle \nabla_z \mathcal{L}(z_t, \lambda_{t+1}) - \hat{\nabla}_z \mathcal{L}(z_t, \lambda_{t+1}), z_{t+1} - z \rangle + \lambda_{t+1}(\tilde{h}(z) - \tilde{h}(z_{t+1})) \\
&+ \frac{\eta_t - \mu}{2}\|z - z_t\|_2^2 - \frac{\eta_t - (\rho_f + \lambda_{t+1}\rho_h)}{2}\|z_{t+1} - z_t\|_2^2 - \frac{\eta_t}{2}\|z - z_{t+1}\|_2^2.
\end{aligned} \tag{49}$$

Recall the definition of $\xi_t = \hat{h}(z_t) - \hat{h}(z_{t-1})$. Substituting it into eq. (42) yields

$$\begin{aligned}
0 \leq{}& -(\lambda - \lambda_{t+1})\hat{h}(z_{t+1}) - (\lambda_{t+1} - \lambda)\xi_{t+1} + \theta_t(\lambda_{t+1} - \lambda)\xi_t \\
&+ \frac{\tau_t}{2}\left((\lambda - \lambda_t)^2 - (\lambda_{t+1} - \lambda_t)^2 - (\lambda - \lambda_{t+1})^2\right).
\end{aligned} \tag{50}$$

Let $w = (z, \lambda)$ and $w_{t+1} = (z_{t+1}, \lambda_{t+1})$. By the definition of the primal-dual gap function, we have

$$\begin{aligned}
&Q(w_{t+1}, w) \\
&= f(z_{t+1}) + \lambda\tilde{h}(z_{t+1}) - f(z) - \lambda_{t+1}\tilde{h}(z) \\
&\overset{(i)}{\leq} \langle \nabla_z \mathcal{L}(z_t, \lambda_{t+1}) - \hat{\nabla}_z \mathcal{L}(z_t, \lambda_{t+1}), z_{t+1} - z \rangle + (\lambda - \lambda_{t+1})\tilde{h}(z_{t+1}) \\
&\quad + \frac{\eta_t - \mu}{2}\|z - z_t\|_2^2 - \frac{\eta_t - (\rho_f + \lambda_{t+1}\rho_h)}{2}\|z_{t+1} - z_t\|_2^2 - \frac{\eta_t}{2}\|z - z_{t+1}\|_2^2. \\
&\overset{(ii)}{\leq} \langle \nabla_z \mathcal{L}(z_t, \lambda_{t+1}) - \hat{\nabla}_z \mathcal{L}(z_t, \lambda_{t+1}), z_{t+1} - z \rangle + (\lambda - \lambda_{t+1})(\tilde{h}(z_{t+1}) - \hat{h}(z_{t+1})) \\
&\quad - (\lambda_{t+1} - \lambda)\xi_{t+1} + \theta_t(\lambda_{t+1} - \lambda)\xi_t + \frac{\tau_t}{2}\left((\lambda - \lambda_t)^2 - (\lambda_{t+1} - \lambda_t)^2 - (\lambda - \lambda_{t+1})^2\right) \\
&\quad + \frac{\eta_t - \mu}{2}\|z - z_t\|_2^2 - \frac{\eta_t - (\rho_f + B\rho_h)}{2}\|z_{t+1} - z_t\|_2^2 - \frac{\eta_t}{2}\|z - z_{t+1}\|_2^2,
\end{aligned} \tag{51}$$

where $(i)$ follows from eq. (49) and $(ii)$ follows from eq. (50) and $0 \leq \lambda_{t+1} \leq B$.

Now we proceed to bound the term $|\tilde{h}(z_t) - \hat{h}(z_t)|$.

$$|\tilde{h}(z_t) - \hat{h}(z_t)| = |g(x_t, \tilde{y}_t^*) - g(x_t, \hat{y}_t^*)| \overset{(i)}{\leq} 2L_g\|\tilde{y}_t^* - \hat{y}_t^*\|_2 \overset{(ii)}{\leq} L_g D_{\mathcal{Z}}\left(1 - \frac{\alpha}{\rho_g + 2\alpha}\right)^N, \tag{52}$$

where $(i)$ follows from Assumption 1, $\mathcal{Z}$ is bounded set, and because we let $L_g := \sup_z \|\nabla_z g(z)\|_2$, and $(ii)$ follows from Lemma 4 and because $\|\hat{y}_0 - \tilde{y}^*(x_t)\|_2 \leq D_{\mathcal{Z}}$.

The following inequality follows immediately from eq. (52) and the fact that $|\lambda - \lambda_{t+1}| \leq B$:

$$(\lambda - \lambda_{t+1})(\tilde{h}(z_t) - \hat{h}(z_t)) \leq |\lambda - \lambda_{t+1}||\tilde{h}(z_t) - \hat{h}(z_t)| \leq L_g B D_{\mathcal{Z}}\left(1 - \frac{\alpha}{\rho_g + 2\alpha}\right)^N. \tag{53}$$

By the definitions of $\nabla_z \mathcal{L}(z_t, \lambda_{t+1})$ and $\hat{\nabla}_z \mathcal{L}(z_t, \lambda_{t+1})$, we have

$$
\begin{aligned}
\|\nabla_z \mathcal{L}(z_t, \lambda_{t+1}) &- \hat{\nabla}_z \mathcal{L}(z_t, \lambda_{t+1})\|_2 \\
&= \left\| \nabla f(z_t) + \lambda_{t+1} \nabla \tilde{h}(z_t) - \left( \nabla f(z_t) + \lambda_{t+1} \hat{\nabla} \tilde{h}(z_t) \right) \right\|_2 \\
&= \lambda_{t+1} \|\nabla g(x_t, \tilde{y}_t^*) - \nabla g(x_t, \hat{y}_t^*)\|_2 \\
&\overset{(i)}{\leq} \lambda_{t+1} \rho_g \|\tilde{y}_t^* - \hat{y}_t^*\|_2 \overset{(ii)}{\leq} B \rho_g D_{\mathcal{Z}} \left( 1 - \frac{\alpha}{\rho_g + 2\alpha} \right)^N,
\end{aligned}
\tag{54}
$$

where $(i)$ follows from Assumption 1 and $(ii)$ follows from Lemma 4, and because $\lambda_{t+1} \leq B$ and $\|\hat{y}_0 - \tilde{y}^*(x_t)\|_2 \leq D_{\mathcal{Z}}$.

By Cauchy-Schwartz inequality and eq. (54), we have

$$
\begin{aligned}
\langle \nabla_z \mathcal{L}(z_t, \lambda_{t+1}) &- \hat{\nabla}_z \mathcal{L}(z_t, \lambda_{t+1}), z_{t+1} - z \rangle \\
&\leq \|\nabla_z \mathcal{L}(z_t, \lambda_{t+1}) - \hat{\nabla}_z \mathcal{L}(z_t, \lambda_{t+1})\|_2 \|z_{t+1} - z\|_2 \leq B \rho_g D_{\mathcal{Z}}^2 \left( 1 - \frac{\alpha}{\rho_g + 2\alpha} \right)^N.
\end{aligned}
\tag{55}
$$

By the definition of $\xi_t$, we have

$$
\begin{aligned}
\theta_t(\lambda_{t+1} - \lambda_t)\xi_t &= \theta_t(\lambda_{t+1} - \lambda_t)(\hat{h}(z_t) - \hat{h}(z_{t-1})) \\
&= \theta_t(\lambda_{t+1} - \lambda_t)(\hat{h}(z_t) - \tilde{h}(z_t) - \hat{h}(z_{t-1}) + \tilde{h}(z_{t-1}) + \tilde{h}(z_t) - \tilde{h}(z_{t-1})) \\
&\leq \theta_t|\lambda_{t+1} - \lambda_t| \left( |\hat{h}(z_t) - \tilde{h}(z_t)| + |\hat{h}(z_{t-1}) - \tilde{h}(z_{t-1})| + |\tilde{h}(z_t) - \tilde{h}(z_{t-1})| \right) \\
&\overset{(i)}{\leq} |\lambda_{t+1} - \lambda_t| \left( 2 L_g D_{\mathcal{Z}} \left( 1 - \frac{\alpha}{\rho_g + 2\alpha} \right)^N + L_g \|z_t - z_{t-1}\|_2 \right) \\
&\overset{(ii)}{\leq} 2 B L_g D_{\mathcal{Z}} \left( 1 - \frac{\alpha}{\rho_g + 2\alpha} \right)^N + L_g |\lambda_{t+1} - \lambda_t| \|z_t - z_{t-1}\|_2 \\
&\overset{(iii)}{\leq} 2 B L_g D_{\mathcal{Z}} \left( 1 - \frac{\alpha}{\rho_g + 2\alpha} \right)^N + \frac{\tau_t}{2}(\lambda_{t+1} - \lambda_t)^2 + \frac{L_g}{2\tau_t}\|z_t - z_{t-1}\|_2^2,
\end{aligned}
\tag{56}
$$

where $(i)$ follows from eq. (52), and because $\theta_t \leq 1$, and $\tilde{h}(z)$ is $L_g$ Lipschitz continuous, $(ii)$ follows because $0 \leq \lambda_t, \lambda_{t+1} \leq B$, and $(iii)$ follows from Young's inequality.

Substituting eqs. (53), (55) and (56) into eq. (51) yields

$$
\begin{aligned}
Q(w_{t+1}, w) \leq &-(\lambda_{t+1} - \lambda)\xi_{t+1} + \theta_t(\lambda_t - \lambda)\xi_t + (\rho_g D_{\mathcal{Z}} + 3 L_g) B D_{\mathcal{Z}} \left( 1 - \frac{\alpha}{\rho_g + 2\alpha} \right)^N \\
&+ \frac{\tau_t}{2} \left( (\lambda - \lambda_t)^2 - (\lambda - \lambda_{t+1})^2 \right) + \frac{\eta_t - \mu}{2}\|z - z_t\|_2^2 - \frac{\eta_t}{2}\|z - z_{t+1}\|_2^2 \\
&+ \frac{L_g^2}{2\tau_t}\|z_t - z_{t-1}\|_2^2 - \frac{\eta_t - (\rho_f + B\rho_h)}{2}\|z_t - z_{t+1}\|_2^2.
\end{aligned}
\tag{57}
$$

Recall that $\gamma_t$, $\theta_t$, $\eta_t$ and $\tau_t$ are set to satisfy $\gamma_{t+1}\theta_{t+1} = \gamma_t$, $\gamma_t \tau_t \geq \gamma_{t+1}\tau_{t+1}$, $\gamma_t \eta_t \geq \gamma_{t+1}(\eta_{t+1} - \mu)$, and

$$
\gamma_t(\rho_f + B\rho_h - \eta_t) + \frac{2\gamma_{t+1} L_g^2}{\tau_{t+1}} \leq 0.
$$

Multiplying $\gamma_t$ on both sides of eq. (57) and telescoping from $t = 0, 1, \ldots T - 1$ yield

$$
\begin{aligned}
\sum_{t=0}^{T-1} \gamma_t Q(w_{t+1}, w) \leq &-\gamma_{T-1}(\lambda_T - \lambda)\xi_T + (\rho_g D_{\mathcal{Z}} + 3 L_g) B D_{\mathcal{Z}} \left( 1 - \frac{\alpha}{\rho_g + 2\alpha} \right)^N \sum_{t=0}^{T-1} \gamma_t \\
&+ \frac{\gamma_0 \tau_0}{2}(\lambda - \lambda_0)^2 + \frac{\gamma_0(\eta_0 - \mu)}{2}\|z - z_0\|_2^2 \\
&- \frac{\gamma_{T-1}(\eta_{T-1} - (\rho_f + B\rho_h))}{2}\|z - z_T\|_2^2.
\end{aligned}
$$

Dividing both sides of the above inequality by $\Gamma_T = \sum_{t=0}^{T-1} \gamma_t$, we obtain

$$\frac{1}{\Gamma_T} \sum_{t=0}^{T-1} \gamma_t Q(w_{t+1}, w) \le - \frac{\gamma_{T-1}(\lambda_T - \lambda)\xi_T}{\Gamma_T} + (\rho_g D_{\mathcal{Z}} + 3L_g) B D_{\mathcal{Z}} \left( 1 - \frac{\alpha}{\rho_g + 2\alpha} \right)^N$$
$$+ \frac{\gamma_0 \tau_0}{2\Gamma_T} (\lambda - \lambda_0)^2 + \frac{\gamma_0(\eta_0 - \mu)}{2\Gamma_T} \|z - z_0\|_2^2$$
$$- \frac{\gamma_{T-1}(\eta_{T-1} - (\rho_f + B\rho_h))}{2\Gamma_T} \|z - z_T\|_2^2. \tag{58}$$

By following the steps similar to those in eq. (56), we have

$$|(\lambda_T - \lambda)\xi_T| \le |\lambda_T - \lambda| \left( 2L_g D_{\mathcal{Z}} \left( 1 - \frac{\alpha}{\rho_g + 2\alpha} \right)^N + L_g \|z_T - z_{T-1}\|_2 \right)$$
$$\le 2L_g B D_{\mathcal{Z}} \left( 1 - \frac{\alpha}{\rho_g + 2\alpha} \right)^N + L_g B D_{\mathcal{Z}}.$$

Recall the definition: $\bar{w} := \frac{1}{\Gamma_T} \sum_{t=0}^{T-1} \gamma_t w_{t+1}$. Noting that $Q(\cdot, w)$ is a convex function and substituting the above inequality into eq. (58) yield

$$Q(\bar{w}, w)$$
$$\le \frac{1}{\Gamma_T} \sum_{t=0}^{T-1} \gamma_t Q(w_{t+1}, w)$$
$$\le \frac{2L_g B D_{\mathcal{Z}}}{\Gamma_T} \left( 1 - \frac{\alpha}{\rho_g + 2\alpha} \right)^N + \frac{L_g B D_{\mathcal{Z}}}{\Gamma_T} + (\rho_g D_{\mathcal{Z}} + 3L_g) B D_{\mathcal{Z}} \left( 1 - \frac{\alpha}{\rho_g + 2\alpha} \right)^N$$
$$+ \frac{\gamma_0 \tau_0}{2\Gamma_T} (\lambda - \lambda_0)^2 + \frac{\gamma_0(\eta_0 - \mu)}{2\Gamma_T} \|z - z_0\|_2^2 - \frac{\gamma_{T-1}(\eta_{T-1} - (\rho_f + B\rho_h))}{2\Gamma_T} \|z - z_T\|_2^2. \tag{59}$$

Let $w = (z^*, 0)$. Then, we have

$$Q(\bar{w}, w) = f(\bar{z}) - f(z^*) - \bar{\lambda}\tilde{h}(z^*) \overset{(i)}{\ge} f(\bar{z}) - f(z^*),$$

where $(i)$ follows from the fact $\tilde{h}(z^*) \le 0$ and $\bar{\lambda} = \frac{1}{\Gamma_T} \sum_{t=0}^{T-1} \gamma_t \lambda_{t+1} \ge 0$.

Substituting the above inequality into eq. (59) yields

$$f(\bar{z}) - f(z^*) \le \frac{2L_g B D_{\mathcal{Z}}}{\Gamma_T} \left( 1 - \frac{\alpha}{\rho_g + 2\alpha} \right)^N + \frac{L_g B D_{\mathcal{Z}}}{\Gamma_T}$$
$$+ (\rho_g D_{\mathcal{Z}} + 3L_g) B D_{\mathcal{Z}} \left( 1 - \frac{\alpha}{\rho_g + 2\alpha} \right)^N + \frac{\gamma_0(\eta_0 - \mu)\|z^* - z_0\|_2^2}{2\Gamma_T}. \tag{60}$$

Recall that $(z^*, \lambda^*)$ is a Nash equilibrium of $\mathcal{L}(z, \lambda)$ and it satisfies $\lambda^* \tilde{h}(z^*) = 0$. Then we have

$$\mathcal{L}(\bar{z}, \lambda^*) \ge \mathcal{L}(z^*, \lambda^*) \overset{\text{by def.}}{\Longleftrightarrow} f(\bar{z}) + \lambda^* \tilde{h}(\bar{z}) - f(z^*) \ge 0. \tag{61}$$

If $\tilde{h}(\bar{z}) \le 0$, the constraint violation $[\tilde{h}(\bar{z})]_+ = 0$, which satisfies the statement in the theorem. If $\tilde{h}(\bar{z}) > 0$, let $w = (z^*, \lambda^* + 1)$. Then, we have

$$Q(\bar{w}, w) = f(\bar{z}) + (\lambda^* + 1)\tilde{h}(\bar{z}) - f(z^*) - \bar{\lambda}\tilde{h}(z^*) \overset{(i)}{\le} f(\bar{z}) + (\lambda^* + 1)\tilde{h}(\bar{z}) - f(z^*), \tag{62}$$

where $(i)$ follows from the facts $\tilde{h}(z^*) \le 0$ and $\bar{\lambda} \ge 0$.

Equations (59), (61) and (62) and the condition $\tilde{h}(\bar{z}) > 0$ together yield,

$$
\begin{aligned}
[\tilde{h}(\bar{z})]_+ = \tilde{h}(\bar{z}) &= Q(\bar{w}, w) - (f(\bar{w}) + \lambda^* \tilde{h}(\bar{z}) - f(z^*)) \le Q(\bar{w}, w) \\
&\le \frac{2L_g B D_{\mathcal{Z}}}{\Gamma_T} \left(1 - \frac{\alpha}{\rho_g + 2\alpha}\right)^N + \frac{L_g B D_{\mathcal{Z}}}{\Gamma_T} + (\rho_g D_{\mathcal{Z}} + 3L_g) B D_{\mathcal{Z}} \left(1 - \frac{\alpha}{\rho_g + 2\alpha}\right)^N \\
&\quad + \frac{\gamma_0 \tau_0}{2\Gamma_T}(\lambda^* + 1)^2 + \frac{\gamma_0(\eta_0 - \mu)}{2\Gamma_T} \|z^*\|_2^2.
\end{aligned}
\tag{63}
$$

Finally, taking $w^* = (z^*, \lambda^*)$ in eq. (59), noticing the fact $Q(w, w^*) \ge 0$ for all $w$, and rearranging the terms, we have

$$
\begin{aligned}
\|\bar{z} - z^*\|_2^2 &\le \frac{1}{\gamma_{T-1}(\eta_{T-1} - (\rho_f + B\rho_h))} \left(4L_g B D_{\mathcal{Z}} \left(1 - \frac{\alpha}{\rho_g + 2\alpha}\right)^N + 2L_g B D_{\mathcal{Z}}\right) \\
&\quad + \frac{1}{\gamma_{T-1}(\eta_{T-1} - (\rho_f + B\rho_h))} \left(\gamma_0 \tau_0 (\lambda^* - \lambda_0)^2 + \gamma_0(\eta_0 - \mu)\|z^* - z_0\|_2^2\right) \\
&\quad + \frac{\Gamma_T}{\gamma_{T-1}(\eta_{T-1} - (\rho_f + B\rho_h))} (\rho_g D_{\mathcal{Z}} + 3L_g) B D_{\mathcal{Z}} \left(1 - \frac{\alpha}{\rho_g + 2\alpha}\right)^N.
\end{aligned}
\tag{64}
$$

Moreover, using the fact that $\Gamma_T \ge \frac{T^2}{2}$ and $\Gamma_T \ge 2$, eq. (60) yields

$$
f(\bar{z}) - f(z^*) \le \frac{2L_g B D_{\mathcal{Z}}}{T^2} + \frac{\gamma_0(\eta_0 - \mu)\|z^* - z_0\|_2^2}{T^2} + (\rho_g D_{\mathcal{Z}} + 4L_g) B D_{\mathcal{Z}} \left(1 - \frac{\alpha}{\rho_g + 2\alpha}\right)^N.
$$

Equation (63) together with the facts that $\Gamma_T \ge 2$, $\Gamma_T \ge T^2/2$, $\|z^*\|_2 \le D_{\mathcal{Z}}$, and $\lambda^* + 1 \le B$, implies

$$
[\tilde{h}(\bar{z})]_+ \le \frac{2L_g B D_{\mathcal{Z}} + \gamma_0 \tau_0 B^2 + \gamma_0(\eta_0 - \mu)D_{\mathcal{Z}}^2}{T^2} + (\rho_g D_{\mathcal{Z}} + 4L_g) B D_{\mathcal{Z}} \left(1 - \frac{\alpha}{\rho_g + 2\alpha}\right)^N.
$$

Using the fact that $\gamma_{T-1}(\eta_{T-1} - \rho_f - B\rho_h) \ge \frac{\mu T^2}{2}$ and $\Gamma_T \le \mu(T + t_0 + 2)^2$, eq. (64) yields

$$
\|\bar{z} - z^*\|_2^2 \le \frac{2\gamma_0 \tau_0 B^2 + 2\gamma_0(\eta_0 - \mu)D_{\mathcal{Z}}^2}{\mu T^2} + \frac{2(T + t_0 + 1)^2}{T^2} (\rho_g D_{\mathcal{Z}} + 4L_g) B D_{\mathcal{Z}} \left(1 - \frac{\alpha}{\rho_g + 2\alpha}\right)^N.
$$

$\square$

## H   Proof of Lemma 1 in Section 4

Recall that we have already shown that $\|\nabla^2 \tilde{g}^*(z)\|_2 \le \rho_g + \frac{\rho_g^2}{\alpha}$ in eq. (41) for all $z \in \mathcal{Z}$. Then, the following inequality holds for all $x, x' \in \mathcal{X}$

$$
\tilde{g}^*(x') \le \tilde{g}^*(x) + \langle \nabla_x \tilde{g}^*(x), x' - x \rangle + \frac{\rho_g + \rho_g^2/\alpha}{2} \|x' - x\|_2^2.
\tag{65}
$$

Recall the inequality in Assumption 1:

$$
g(x', y') \ge g(x, y) + \langle \nabla_x g(x, y), x' - x \rangle + \langle \nabla_y g(x, y), y' - y \rangle - \frac{\rho_g}{2} \|x - x'\|_2^2.
\tag{66}
$$

Consider $\tilde{h}_k(z)$. We have $\nabla_z \tilde{h}_k(z) = \nabla_z g(z) - (\nabla_x \tilde{g}^*(x); \mathbf{0}_d) + (2\rho_g + \frac{\rho_g^2}{\alpha})(x - \tilde{x}_{k-1}; \mathbf{0}_d)$. Then, we have

$$
\begin{aligned}
&\tilde{h}_k(z) + \langle \nabla_z \tilde{h}_k(z), z' - z \rangle \\
&= g(x, y) - \tilde{g}^*(x) + \frac{(2\rho_g + \frac{\rho_g^2}{\alpha})}{2} \|x - \tilde{x}_{k-1}\|_2^2 - \delta + \langle \nabla_x g(x, y), x' - x \rangle - \langle \nabla_x \tilde{g}^*(x), x' - x \rangle \\
&\quad + \langle \nabla_y g(x, y), y' - y \rangle + (2\rho_g + \frac{\rho_g^2}{\alpha})\langle x - \tilde{x}_{k-1}, x' - x \rangle.
\end{aligned}
\tag{67}
$$

Substituting eqs. (65) and (66) into eq. (67), we obtain

$$
\tilde{h}_k(z) + \langle \nabla_z \tilde{h}_k(z), z' - z \rangle \le g(x', y') - \tilde{g}(x') + \frac{2\rho_g + \rho_g^2/\alpha}{2} \|x' - \tilde{x}_{k-1}\|_2^2 - \delta = \tilde{h}_k(z').
$$

The above inequality is the necessary and sufficient condition for convexity, which completes the proof.

# I  Proof of Theorem 2

We first formally restate the theorem with the full details.

**Theorem 4** (Formal Statement of Theorem 2). *Suppose that Assumption 1 holds. Consider Algorithm 2. Let the hyperparameters $B > 0$ be a large enough constant, $\gamma_t = t + t_0 + 1$, $\eta_t = \frac{\rho_f(t+t_0+1)}{2}$, $\tau_t = \frac{4L_g^2}{\rho_f t}$, $\theta_t = \frac{t+t_0}{t+t_0+1}$, where $t_0 = \frac{6\rho_f + 4B\rho_h}{\rho_f}$, $\rho_h$ is specified in Lemma 6, and $L_g = \sup_{z \in \mathcal{Z}} \|\nabla g(z)\|_2$. Set $B = \frac{D_f + \rho_f D_{\mathcal{Z}}^2}{\delta} + 1$, where $D_f = \sup_{z,z' \in \mathcal{Z}} |f(z) - f(z')|$. Then, the output $\tilde{z}_{\hat{k}}$ of Algorithm 2 with a randomly chosen index $\hat{k}$ is a stochastic $\epsilon$-KKT point of eq. (2), where $\epsilon$ is given by $\epsilon = \mathcal{O}\left(\frac{1}{K}\right) + \mathcal{O}\left(\frac{1}{T^2}\right) + \mathcal{O}\left(e^{-N}\right)$.*

Central to the proof of Theorem 2, we first prove the uniform bound of the optimal dual variables as stated in the following lemma.

**Lemma 7.** *For each subproblem $(P_k)$, there exists a unique global optimizer $z_k^*$ and optimal dual variable $\lambda_k^*$ such that $\lambda_k^* \leq \bar{B} := (D_f + \rho_f D_{\mathcal{Z}}^2)/\delta$, where $D_f := \sup_{z,z' \in \mathcal{Z}} |f(z) - f(z')|$.*

*Proof of Lemma 7.* For each subproblem $(P_k)$, let $\bar{z}_{k-1} = (\tilde{x}_{k-1}, \bar{y}_{k-1})$ with $\bar{y}_{k-1} = \arg\min_{\mathbb{R}^d} g(\tilde{x}_{k-1}, y) + \frac{\alpha\|y\|_2^2}{2}$. Then, we have $\tilde{h}_k(\bar{z}_{k-1}) = -\delta$, which is a strictly feasible point.

Define the function $d_k(\lambda) = \min_{z \in \mathcal{Z}} \mathcal{L}_k(z, \lambda)$. Then, for any $\lambda$ and $z \in \mathcal{Z}$, we have

$$d_k(\lambda) \leq f_k(\bar{z}_{k-1}) + \lambda \tilde{h}_k(\bar{z}_{k-1}) = f_k(\bar{z}_{k-1}) - \delta\lambda. \tag{68}$$

Moreover, it is known that constrained optimization with strongly convex objective and strongly convex constraints has no duality gap. Combining this with the fact that the strictly feasible point exists, we conclude that the optimal dual variable exists in $\mathbb{R}_+$. Taking $\lambda = \lambda_k^*$ in eq. (68) and using the fact that $|d_k(\lambda^*) - f_k(\bar{z}_{k-1})| = |f_k(z_k^*) - f_k(\bar{z}_{k-1})| \leq D_f + \rho_f D_{\mathcal{Z}}^2$, we complete the proof. $\square$

To proceed the proof of Theorem 2, the function $f_k(z)$ in the subproblem $(P_k)$ is a $\mu = \rho_f$ strongly convex and $3\rho_f$ Lipschitz continuous function, and $h_k(z)$ is a convex function and $2\rho_h$ Lipschitz continuous function. Thus, as we state in the theorem, let $\gamma_t$, $\tau_t$, $\theta_t$ and $\eta_t$ be the same as those in Theorem 3 with $\mu = \rho_f$, $\rho_f$ being replaced by $3\rho_f$, and $\rho_h$ being replaced by $2\rho_h$. We then follow steps similar to those in the proof of Theorem 3. In particular, note that Lemma 7 indicates that for each subproblem $(P_k)$, the optimal dual variable exists and is bounded by $\bar{B} = (D_f + \rho_f D_{\mathcal{Z}}^2)/\delta$. Thus, setting $\Lambda = [0, B]$ with $B = \bar{B} + 1$ ensures both $\lambda_k^*$ and $\lambda_k^* + 1$ to be inside the set $\Lambda$, which further ensures that we can follow steps similar to eqs. (62) and (64). We then obtain for all $k \in \mathbb{N}$, the following bounds hold:

$$f_k(\tilde{z}_k) - f_k(z_k^*) \leq \mathcal{O}\left(\frac{1}{T^2}\right) + \mathcal{O}\left(e^{-N}\right),$$

$$[\tilde{h}_k(\tilde{z}_k)]_+ \leq \mathcal{O}\left(\frac{1}{T^2}\right) + \mathcal{O}\left(e^{-N}\right),$$

and

$$\|\tilde{z}_k - z_k^*\|_2^2 \leq \mathcal{O}\left(\frac{1}{T^2}\right) + \mathcal{O}\left(e^{-N}\right).$$

With the above convergence rate bounds, we apply the following result on the convergence of the nonconvex constrained problem.

**Lemma 8** (Theorem 3.17 Boob et al. (2019)). *Suppose Assumption 1 hold. Denote the global optimizer of $(P_k)$ as $z_k^*$. Suppose the optimal dual variable has an upper bound $\bar{B}$, each subproblem is solved to $\Delta$-accuracy, i.e., the optimality gap $f_k(\tilde{z}_k) - f_k(z_k^*) \leq \Delta$, constraint violation $[\tilde{h}_k(\tilde{z}_k)]_+ \leq \Delta$, and distance to the solution $\|\tilde{z}_k - z_k^*\|_2^2 \leq \Delta$. Then, the final output with a randomly chosen index is an stochastic $\epsilon$-KKT point of eq. (3), with $\epsilon = \frac{D_f + \rho_f D_{\mathcal{Z}}^2 + \bar{B}V_0}{K} + \frac{16(\bar{B}+1)}{\rho_f}\Delta$, where $D_f = \sup_{z,z' \in \mathcal{Z}} |f(z) - f(z')|$, $V_0 = \max\{\tilde{h}(\tilde{z}_0), 0\}$.*

Applying Lemma 8 with $\Delta = \mathcal{O}(\frac{1}{T^2}) + \mathcal{O}(e^{-N})$, we conclude that $\tilde{z}_{\hat{k}}$ is an $\epsilon$-KKT point of eq. (3) where $\epsilon$ is specified below as

$$\epsilon = \mathcal{O}\left(\frac{1}{K}\right) + \mathcal{O}\left(\frac{1}{T^2}\right) + \mathcal{O}(e^{-N}).$$

This completes the proof.

