# OpenReview forum: "A Primal-Dual Approach to Bilevel Optimization with Multiple Inner Minima"
_TMLR — Rejected by TMLR_

### Review · Reviewer_vpcW · 2024-04-03

**Summary Of Contributions:**

This paper studies the bilevel optimization problem with multiple minima at the lower level where it proposes a primal dual algorithm for tackling the problem. The algorithm is developed through (a) reformulating the lower level problem into an inequality constrained problem in (2), which is then further approximated by (3), (b) considering the dual problem of (2) and applying a primal-dual optimizer (with acceleration). Further, the proposed algorithm is shown to enjoy a sublinear convergence rate of $O(1/T^2)$ under certain assumption.

**Audience:**

Yes

**Broader Impact Concerns:**

None.

**Claims And Evidence:**

Yes

**Requested Changes:**

See the above listed weaknesses/shortcomings.

**Strengths And Weaknesses:**

Overall, the reviewer finds the claims in the paper to be correct and reasonable. Meanwhile, there are a number of shortcomings in the current manuscript:

1. The paper reads like a lightly edited resubmission, which is evidenced by the lack of discussions with prior works that appear/are published after 2022. In fact, the paper only provides a very short discussion to compare the paper to recent works published after 2022 which *address the same/similar class of bilevel optimization problem* as the current paper. Given the similarity to these works, it is essential to make a thorough comparison, including both the methodology, convergence results, and numerical experiments.

On a minor note, the reviewer finds the sentence "After this work was initially posted on arXiv, ..." to be slightly inappropriate, given the double blind review policy of TMLR.

2. In the development between (2) to (3), the paper proposes a relaxation of the constraint in (2) through using a smooth term of $\tilde{g}^\star(x)$ that adds a squared norm regularization $||y||^2$ to the lower level subproblem, the reviewer wonders if such perturbation together with the $\delta$ modification in (3) may turn the optimal solution to (3) into one that is not the same as (1). From the statement of Lemma 5, it seems the solution denoted as $z^\star$ in this paper actually refers to the optimal solution of (3) only. If this is the case, it will be important to also quantify (analytically) the distance between $z^\star$ and the optimal solution of (1).

Furthermore, as one of the key issue tackled in this work is to work with lower level problem that is not strongly convex, the reviewer wonders if one can also apply a similar trick to (3) to approximate the lower level problem by a strongly convex one. In the latter case, it seems that one can apply the other existing bilevel algorithm to tackle the same problem.

3. In assumption 2, the paper requires that the function $f(z)$ is strongly convex in $z$ and $\tilde{h}(z)$ is convex in $z$, i.e., the joint variable. The assumption seems to be quite strong. The reviewer wonders if the paper could provide some concrete example where the said assumption can be satisfied.

---

> ### Author Response · Authors · 2024-05-05
>
> Thank you for your thorough reviews and constructive comments. We provide our response to your comments below.
>
> Q1: The paper reads like a lightly edited resubmission, which is evidenced by the lack of discussions with prior works that appear/are published after 2022. In fact, the paper only provides a very short discussion to compare the paper to recent works published after 2022 which address the same/similar class of bilevel optimization problem as the current paper. Given the similarity to these works, it is essential to make a thorough comparison, including both the methodology, convergence results, and numerical experiments.
>
> A1: Thanks for the suggestions! We have added more recent baselines (after 2022) in our comparisons table in the Appendix. However, we found numerical comparisons with those new baselines to be unfair because these methods are either based on new techniques developed after our paper appeared or use techniques that are based on our work.
>
> Q2: On a minor note, the reviewer finds the sentence "After this work was initially posted on arXiv, ..." to be slightly inappropriate, given the double blind review policy of TMLR.
>
> A2: We understand the reviewer’s concern, however this is a known trick in the community that can help to put the paper into the right context. Otherwise, it could be hard for some reviewers to see the true value of the paper by comparing it to more recent works that appeared after the paper.
>
> Q3: In the development between (2) to (3), the paper proposes a relaxation of the constraint in (2) through using a smooth term of $\tilde g^*(x)$ that adds a squared norm regularization $||y||^2$ to the lower level subproblem, the reviewer wonders if such perturbation together with the modification in (3) may turn the optimal solution to (3) into one that is not the same as (1). From the statement of Lemma 5, it seems the solution denoted as $z^*$ in this paper actually refers to the optimal solution of (3) only. If this is the case, it will be important to also quantify (analytically) the distance between $z^*$ and the optimal solution of (1).
> Furthermore, as one of the key issue tackled in this work is to work with lower level problem that is not strongly convex, the reviewer wonders if one can also apply a similar trick to (3) to approximate the lower level problem by a strongly convex one. In the latter case, it seems that one can apply the other existing bilevel algorithm to tackle the same problem.
>
> A3: Thank you for pointing this out! In Appendix F, we have discussed the relationship between the solutions obtained from eqs.(2) and (3). And it turns out that the gap between these eqs. can be upper-bounded in Proposition 1. We have also compared other SOTA results with ours in Appendix A. As shown in Table 2, we can use other methods to tackle each individual reformulated subproblem in practice without non-asymptotic convergence.
>
> Q4: In assumption 2, the paper requires that the function $f$  is strongly convex in $z$ and $h$ is convex in $z$, i.e., the joint variable. The assumption seems to be quite strong. The reviewer wonders if the paper could provide some concrete example where the said assumption can be satisfied.
>
> A4: Thanks for the question! Note that Assumption 2 is for the convergence guarantee of the PDBO algorithm, which is a simplified version under the convex case of our approach. However, we call the reviewer's attention that in Section 4, we further provide Proximal-PDBO, which is more generally applicable to $\textbf{nonconvex}$ $f(x)$ and $\textbf{nonconvex}$ $\tilde{h}(z)$. Theorem 2 in Section 4.2 provides the convergence rate guarantee for such a nonconvex case.

---

### Review · Reviewer_BzVh · 2024-04-08

**Summary Of Contributions:**

This work proposed a new algorithm for bilevel optimization with a convex lower-level problem. The algorithm is based on applying primal-dual algorithm on the smoothed Lagrangian reformulation of the original bilevel problem. Convergence rates under strongly convex and nonconvex upper-level problems are provided.

**Audience:**

Yes

**Broader Impact Concerns:**

/

**Claims And Evidence:**

No

**Requested Changes:**

1. Eq 5, authors mentioned it is a projected GD, but the iterate has no projection
2. Parameter setting of B is not specified in the experiments, you mentioned in the theory that it should be large enough, how do you set it in the experiments?

**Strengths And Weaknesses:**

Strength:
1. Provide non-asymptotic convergence guarantee for bilevel optimization with convex lower-level.
2. The writing of the paper is easy to follow.

Weakness:
1. Rationality of assumptions. The inequality in Assumption 1 and the convexity of $\tilde{h}$ are not fully rationalized. The inequality looks a bit unintuitive to me, in fact with convexity and smoothness, you can derive some similar inequality, I may view this one for technical convenience. Also as authors mentioned, the reformulated problem (Eq. 3) has nonconvex constraint, while the incurred Assumption 2 on $\tilde{h}$ is not fully rationalized, I think it would be better to give some examples showing the convexity really holds.
2. Also authors mentioned the constraint $\mathcal{X}$ is bounded, but the experiments seems to be unconstrained. It may be understandable if the bounded assumption is required for theoretical convenience, but I suggest authors add related discussion for clarification.
3. Lack of related literature. Seems like there are some closely related works on bilevel optimization with multiple inner minimal, e.g., *Chen, Lesi, Jing Xu, and Jingzhao Zhang. "Bilevel Optimization without Lower-Level Strong Convexity from the Hyper-Objective Perspective." arXiv preprint arXiv:2301.00712 (2023)*, also they seem to have some arguments on the value function approach that the authors used. So I suggest authors to add it to your discussion.
4. Experiments. In your experiment results plots, many of them do not share the same starting point (e.g., Fig 1 a,c,d), or the plots show the iterate does not start from the 0 time (e.g., Fig 3 left). It may be better to further polish the experiment results or attach more illustration for such mismatch.

---

> ### Author Response · Authors · 2024-05-05
>
> Thank you for your thorough reviews and constructive comments. We provide our response to your comments below.
>
> Q1: Rationality of assumptions. The inequality in Assumption 1 and the convexity of $\tilde h$ are not fully rationalized. The inequality looks a bit unintuitive to me, in fact with convexity and smoothness, you can derive some similar inequality, I may view this one for technical convenience. Also as authors mentioned, the reformulated problem (Eq. 3) has nonconvex constraint, while the incurred Assumption 2 on is not fully rationalized, I think it would be better to give some examples showing the convexity really holds.
>
> A1: Thanks for the comments! The inequality in Assumption 1 combines two standard assumptions of gradient Lipschitz continuous condition of $g(x,y)$ and the convexity of $g(x, y)$ on $y$ for any $x\in\mathcal{X}$. Fix $x=x^\prime$, it equals the sufficient and necessary geometric characterization of convexity of $g$ on $y$. Fix $y = y^\prime$, it provides the geometric equivalent of gradient Lipschitz continuity. Both assumptions are standard in the bilevel optimization literature.
> For the convexity of $\tilde h$, note that Assumption 2 is for the convergence guarantee of the PDBO algorithm, which is a simplified version under the convex case of our approach. However, we call the reviewer's attention that in Section 4, we further provide Proximal-PDBO, which is more generally applicable to $\textbf{nonconvex}$ $f(x)$ and $\textbf{nonconvex}$ $\tilde{h}(z)$. Theorem 2 in Section 4.2 provides the convergence rate guarantee for such a nonconvex case.
>
> Q2: Also authors mentioned the constraint $\mathcal{X}$ is bounded, but the experiments seems to be unconstrained. It may be understandable if the bounded assumption is required for theoretical convenience, but I suggest authors add related discussion for clarification.
>
> A2: We thank the reviewer for noticing this! For the experiments, it suffices to restrict $\mathcal{X}$ as a bounded subset of  $\mathbb{R}^n$ to ensure the boundedness which we adopted for theoretical convenience. We have clarified this in our revision.
>
> Q3: Lack of related literature. Seems like there are some closely related works on bilevel optimization with multiple inner minimal, e.g., Chen, Lesi, Jing Xu, and Jingzhao Zhang. "Bilevel Optimization without Lower-Level Strong Convexity from the Hyper-Objective Perspective." arXiv preprint arXiv:2301.00712 (2023), also they seem to have some arguments on the value function approach that the authors used. So I suggest authors to add it to your discussion.
>
> A3: Thanks for the suggested reference! This is in fact a very useful reference and we have added a discussion on it in our revision. Note that we provided the discussion on the more recent papers at the end of our related work section.
>
> Q4: Experiments. In your experiment results plots, many of them do not share the same starting point (e.g., Fig 1 a,c,d), or the plots show the iterate does not start from the 0 time (e.g., Fig 3 left). It may be better to further polish the experiment results or attach more illustration for such mismatch.
>
> A4: Thanks for the comment! For the figures, we plot the different quantities after obtaining the first iterate in order to have a more clear view of the compared plots (i.e. a better zoom in because the initial gap is much bigger than the remaining iterates). We also did not plot starting from time 0 in Figure 3 for similar reasons (the initial loss was too big so we plot the iterates with loss less than 200 for each algorithm).
>
> Q5: Eq 5, authors mentioned it is a projected GD, but the iterate has no projection
>
> A5: Thank you for pointing this out. The right hand side of Eq. 5 should be projected to a set $\mathcal{Z}$ that contains all local minima. And because such a set is usually large enough, the projected gradient descent usually equals the gradient descent.
>
> Q6: Parameter setting of B is not specified in the experiments, you mentioned in the theory that it should be large enough, how do you set it in the experiments?
>
> A6: Thanks for the question! We set B to be 100 in all our experiments, but we found that the value of $\lambda$ usually stays within [0, 10] which makes the projection step unnecessary in our experiments. We have specified this in our revision.

---

### Review · Reviewer_UEC5 · 2024-04-22

**Summary Of Contributions:**

In this paper, they tackle the challenge bilevel problem with multiple inner minima. The authors reformulate the bilevel optimization problem and propose a novel primal-dual method (Proximal-DBO) to solve it. The convergence rate analysis for it is provided when the upper-level function is smooth and the lower function is smooth and convex. Experiments have also been done to demonstrate the efficiency of the proposed algorithm.

**Audience:**

Yes

**Claims And Evidence:**

Yes

**Requested Changes:**

The following is my suggestions and please see whether it would be better to present your paper as follows:

1. *Incorporate Proposition 1 into the main content*: Move Proposition 1 into the main body of the text to emphasize which stationary point the algorithm achieved.

2. *Boundedness of $\lambda$*: The $\lambda$ is not bounded if no assumption is imposed. Therefore, in section 3.1, I think you cannot regard $\lambda$ as a bounded variable. I think you should state it as an assumption when introducing your algorithm and prove it always holds in all settings you considered.

3. *Lemma for Calculation of $\nabla \tilde{h} (z)$*: Consider writing a lemma detailing the calculation of $\nabla \tilde{h} (z)$ in the main content, rather than relegating it to an appendix. Rename Appendix E as the proof of this lemma.

**Strengths And Weaknesses:**

# Strengths:

1. This paper is well-written and easy to follow.

2. They propose a new algorithm for solving bilevel optimization with multiple inner minima and provide the first known non-asymptotic convergence guarantee for this problem.

# Weakness:
Here are some questions. Please correct me if I am wrong.

1. In Proposition 1, what are $D$ and $D_z$. From my understanding, they are the upper bound for $\lambda$ and $z$. However, would it be possible to guarantee the boundedness of them just under Assumption 1? I think additional conditions are needed.

2. How to choose $\alpha$ and $\delta$ in your algorithm? I think it would be better to point out the choices for them in your main theorem. BTW, how to choose them in practice? I think you should also mention it in your experiment's parts.

3. In the algorithm Proximal-PDBO, I am wondering whether there is a typo in the definition of $f_k(z)$. Under Assumption 1, why is $f_k (z)$ strongly convex with $\mu=\rho$?

4. In the proof of Lemma 6, why can you use the second-order derivative of $g$? You do not assume it in Assumption 1.

---

> ### Author Response · Authors · 2024-05-05
>
> Thank you for your thorough reviews and constructive comments. We provide our response to your comments below.
>
> Q1:In Proposition 1, what are $D$  and $D_z$. From my understanding, they are the upper bound for $\lambda$ and $z$. However, would it be possible to guarantee the boundedness of them just under Assumption 1? I think additional conditions are needed.
>
> A1: Thank you for pointing this out! For the upper bound of $\lambda$, it may be derived from strictly feasibility of the $g(x, y) - g^*_\alpha(x)$. For the boundness of the $z$, it depends on the union of $\mathcal{S}_x$ over $\mathcal{X}$ and the diameter of set $\mathcal{X}$. They should both be specified explicitly.
>
>
> Q2: How to choose $\alpha$ and $\delta$ in your algorithm? I think it would be better to point out the choices for them in your main theorem. BTW, how to choose them in practice? I think you should also mention it in your experiment's parts.
>
> A2: From the theoretical perspective, the discrepancy of the reformulated problem and the original problem can be upper-bounded by $\mathcal{O}(\epsilon)$, as shown in Proposition 1 in Appendix F. Thus, to have a $\mathcal{O}(\epsilon)$ result of the original problem, we require both $\alpha$ and $\delta$ to be $\mathcal{O}(\epsilon)$.
> From an experimental perspective, we did not find our algorithm to be particularly sensitive to the choices of $\alpha$ and $\delta$ as long as they are small enough. For $\delta$, small values less than 0.1 work well in all our experiments (we fixed it to 0.01). For $\alpha$, we set it to 0.01 in the numerical problem and to 0.001 in all other experiments.
>
>
> Q3: In the algorithm Proximal-PDBO, I am wondering whether there is a typo in the definition of $f_k(z)$. Under Assumption 1, why is $f_k(z)$  strongly convex with $\mu=\rho$?
>
> A3: Under Assumption 1, $f_k(z)$ is $\rho_f$-smooth. Hence, adding the regularizer $\rho_f \||z - \tilde{z}_{k-1}\||_2^2$ will make it $\mu$-strongly convex with $\mu=\rho_f$. We stated this fact at the beginning of section 4.1 and refer the reviewer to, e.g., Boob et al., 2019 for the derivation of this result.
>
> Q4: In the proof of Lemma 6, why can you use the second-order derivative of $g$? You do not assume it in Assumption 1.
>
> A4: We thank the reviewer for noticing this! In fact, we forgot to clearly mention that the function $g$ is twice differentiable, but we’ve now stated it in Assumption 1 of our revision. In this case, $\rho$-smoothness is the same as the second-order gradient being upper bounded by $\rho$, which is what we use in the proof of Lemma 6.
>
> Q5: Requested presentation changes.
>
> A5: We have made the adequate changes. See our answer A1 for the boundedness of $\lambda$. Note that because we have a paragraph in section 3.1 that explains how to obtain the gradient $\nabla \tilde{h}(z)$, we thought it would be redundant to add a lemma for it again. The full proof of the expression of the gradient $\nabla \tilde{h}(z)$ can be found in appendix E.

---

> > ### Comment · Reviewer_UEC5 · 2024-05-10
> > **Proof of Lemma 6 (Theorem 1) without assuming twice differentiability**
> >
> > Thanks for your explaination.
> >
> > For Q4, you assume that the function $g$ is twice differentiable. It is worth noting that your algorithm utilizes only first-order information, yet the convergence proof relies on the assumption of twice differentiability. I think it would be better if the proof did not necessitate second-order differentiability. Therefore, I am curious whether you could prove Lemma 6 and Theorem 1 without assuming twice differentiability.

---

### Decision · Action_Editor_Enct · 2024-06-23

**Recommendation:** Reject

**Comment:**

The paper proposed a new primal-dual bilevel optimization (PDBO) algorithm, which can address "multiple inner minima challenge". I think the paper makes an interesting contribution to the area. However, reviewers menteiond that the paper's assumptions shall be further rationalized, and some technical issues in the claims (e.g., projection) should be further clarified. In addition, it is necessary for the authors to provide an up-to-date account for works published prior to 2024. I think the authors may consider resubmitting the paper after modifying the paper to address the aforementioned issues.

**Audience:**

Yes, the bilevel optimization problem is of interest to many researchers in machine learning area.

**Claims And Evidence:**

As a reviewer pointed out, it is necessary for the authors to provide an up-to-date account for works published prior to 2024. Without the context, it may be hard for reviewers to judge the claims of the paper.

**Resubmission Of Major Revision:**

The authors may consider submitting a major revision at a later time.